# Cyclical Stochastic Gradient MCMC for Bayesian Deep Learning

**Ruqi Zhang**
Cornell University
rz297@cornell.edu

**Chunyuan Li**
Microsoft Research, Redmond
chunyl@microsoft.com

**Jianyi Zhang**
Duke University
jz318@duke.edu

**Changyou Chen**
University at Buffalo, SUNY
changyou@buffalo.edu

**Andrew Gordon Wilson**
New York University
andrewgw@cims.nyu.edu

## Abstract

The posteriors over neural network weights are high dimensional and multimodal. Each mode typically characterizes a meaningfully different representation of the data. We develop Cyclical Stochastic Gradient MCMC (SG-MCMC) to automatically explore such distributions. In particular, we propose a cyclical stepsize schedule, where larger steps discover new modes, and smaller steps characterize each mode. We prove non-asymptotic convergence of our proposed algorithm. Moreover, we provide extensive experimental results, including ImageNet, to demonstrate the effectiveness of cyclical SG-MCMC in learning complex multimodal distributions, especially for fully Bayesian inference with modern deep neural networks.

## 1 Introduction

Deep neural networks are often trained with stochastic optimization methods such as stochastic gradient decent (SGD) and its variants. Bayesian methods provide a principled alternative, which account for model uncertainty in weight space (MacKay, 1992; Neal, 1996), and achieve an automatic balance between model complexity and data fitting. Indeed, Bayesian methods have been shown to improve the generalization performance of DNNs (Hernández-Lobato & Adams, 2015; Blundell et al., 2015; Li et al., 2016a; Maddox et al., 2019), while providing a principled representation of uncertainty on predictions which is crucial for decision making.

Approximate inference for Bayesian deep learning has typically focused on deterministic approaches, such as variational methods (Hernández-Lobato & Adams, 2015; Blundell et al., 2015). By contrast, MCMC methods are now essentially unused for inference with modern deep neural networks, despite previously providing the gold standard of performance with smaller neural networks (Neal, 1996). Stochastic gradient Markov Chain Monte Carlo (SG-MCMC) methods (Welling & Teh, 2011; Chen et al., 2014; Ding et al., 2014; Li et al., 2016a) provide a promising direction for a sampling based approach to inference in Bayesian deep learning. Indeed, it has been shown that stochastic methods, which use mini-batches of data, are crucial for finding weight parameters that provide good generalization in modern deep neural networks (Keskar et al., 2016).

However, SG-MCMC algorithms for inference with modern neural networks face several challenges: (*i*) In theory, SG-MCMC asymptotically converges to target distributions via a decreasing stepsize scheme, but suffers from a bounded estimation error in limited time (Teh et al., 2016; Chen et al., 2015). (*ii*) In practice, empirical successes have been reported by training DNNs in relatively short time (Li et al., 2016b; Chen et al., 2014; Gan et al., 2016; Neelakantan et al., 2016; Saatchi & Wilson, 2017). For example, Saatchi & Wilson (2017) apply SG-MCMC to generative adversarial networks (GANs) to solve the mode collapse problem and capture diverse generation styles. However, the loss surface for DNNs is highly multimodal (Auer et al., 1996; Choromanska et al., 2015). In order for MCMC to be effective for posterior inference in modern neural networks, a crucial question remains: how do we make SG-MCMC efficiently explore a highly multimodal parameter space given a practical computational budget?

Several attempts have been made to improve the sampling efficiency of SG-MCMC. Stochastic Gradient Hamiltonian Monte Carlo (SGHMC) (Chen et al., 2014) introduces momentum to Langevin dynamics. Preconditioned stochastic gradient Langevin dynamics (pSGLD) (Li et al., 2016a) adaptively adjusts the sampler's step size according to the local geometry of parameter space. Though simple and promising, these methods are still inefficient at exploring multimodal distributions in practice. It is our contention that this limitation arises from difficulties escaping local modes when using the small stepsizes that SG-MCMC methods typically require. Note that the stepsize in SG-MCMC controls the sampler's behavior in two ways: the magnitude to deterministically drift towards high density regions *wrt.* the current stochastic gradient, and the level of injecting noise to randomly explore the parameter space. Therefore, a small stepsize reduces both abilities, resulting in a large numbers of iterations for the sampler to move across the modes.

In this paper, we propose to replace the traditional decreasing stepsize schedule in SG-MCMC with a cyclical variant. To note the distinction from traditional SG-MCMC, we refer to this method as *Cyclical SG-MCMC* (cSG-MCMC). The comparison is illustrated in Figure 1. The blue curve is the traditional decay, while the red curve shows the proposed cyclical schedule. Cyclical SG-MCMC operates in two stages: (*i*) *Exploration*: when the stepsize is large (dashed red curves), we consider this stage as an effective burn-in mechanism, encouraging the sampler to take large

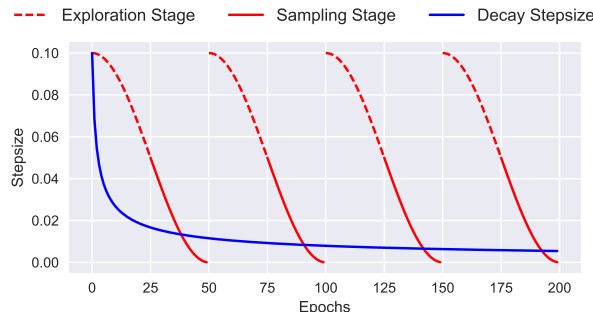

Figure 1: Illustration of the proposed cyclical stepsize schedule (red) and the traditional decreasing stepsize schedule (blue) for SG-MCMC algorithms.

moves and leave the local mode using the stochastic gradient. (*ii*) *Sampling*: when the stepsize is small (solid red curves), the sampler explores one local mode. We collect samples for local distribution estimation during this stage. Further, we propose two practical techniques to improve estimation efficiency: (1) a system temperature for exploration and exploitation; (2) A weighted combination scheme for samples collected in different cycles to reflect their relative importance.

This procedure can be viewed as SG-MCMC with warm restarts: the exploration stage provides the warm restarts for its following sampling stage. cSG-MCMC combines the advantages from (1) the traditional SG-MCMC to characterize the fine-scale local density of a distribution and (2) the cyclical schedule in optimization to efficiently explore multimodal posterior distributions of the parameter space. In limited time, cSG-MCMC is a practical tool to provide significantly better mixing than the traditional SG-MCMC for complex distributions. cSG-MCMC can also be considered as an *efficient* approximation to parallel MCMC; cSG-MCMC can achieve similar performance to parallel MCMC with only a fraction of cost (reciprocal to the number of chains) that parallel MCMC requires.

To support our proposal, we also prove the non-asymptotic convergence for the cyclical schedule. We note that this is the first convergence analysis of a cyclical stepsize algorithm (including work in optimization). Moreover, we provide extensive experimental results to demonstrate the advantages of cSG-MCMC in sampling from multimodal distributions, including Bayesian neural networks and uncertainty estimation on several large and challenging datasets such as ImageNet.

In short, cSG-MCMC provides a simple and automatic approach to inference in modern Bayesian deep learning, with promising results, and theoretical support. This work is a step towards enabling MCMC approaches in Bayesian deep learning. We release code at
https://github.com/ruqizhang/csgmcmc.

## 2 PRELIMINARIES: SG-MCMC WITH A DECREASING STEPSIZE SCHEDULE

SG-MCMC is a family of scalable sampling methods that enables inference with mini-batches of data. For a dataset $\mathcal{D} = \{d_i\}_{i=1}^N$ and a $\theta$-parameterized model, we have the likelihood $p(\mathcal{D}|\theta)$ and prior $p(\theta)$. The posterior distribution is $p(\theta|\mathcal{D}) \propto \exp(-U(\theta))$, where $U(\theta)$ is the potential energy given by $U(\theta) = -\log p(\mathcal{D}|\theta) - \log p(\theta)$.

When $\mathcal{D}$ is too large, it is expensive to evaluate $U(\theta)$ for all the data points at each iteration. Instead, SG-MCMC methods use a minibatch to approximate $U(\theta)$: $\tilde{U}(\theta) = -\frac{N'}{N}\sum_{i=1}^{N'}\log p(x_i|\theta) - \log p(\theta)$, where $N' \ll N$ is the size of minibatch. We recommend Ma et al. (2015) for a general review of SG-MCMC algorithms. We describe two SG-MCMC algorithms considered in this paper.

**SGLD & SGHMC** Welling & Teh (2011) proposed Stochastic Gradient Langevin Dynamics (SGLD), which uses stochastic gradients with Gaussian noise. Posterior samples are updated at the $k$-th step as: $\theta_k = \theta_{k-1} - \alpha_k\nabla\tilde{U}(\theta_k) + \sqrt{2\alpha_k}\epsilon_k$, where $\alpha_k$ is the stepsize and $\epsilon_k$ has a standard Gaussian distribution.

To improve mixing over SGLD, Stochastic Gradient Hamiltonian Monte Carlo (SGHMC) (Chen et al., 2014) introduces an auxiliary momentum variable $v$. SGHMC is built upon HMC, with an additional friction term to counteract the noise introduced by a mini-batch. The update rule for posterior samples is: $\theta_k = \theta_{k-1} + v_{k-1}$, and $v_k = v_{k-1} - \alpha_k\nabla\tilde{U}(\theta_k) - \eta v_{k-1} + \sqrt{2(\eta - \hat{\gamma})\alpha_k}\epsilon_k$, where $1 - \eta$ is the momentum term and $\hat{\gamma}$ is the estimate of the noise.

To guarantee asymptotic consistency with the true distribution, SG-MCMC requires that the step sizes satisfy the following assumption:

**Assumption 1.** *The step sizes $\{\alpha_k\}$ are decreasing, i.e., $0 < \alpha_{k+1} < \alpha_k$, with 1) $\sum_{k=1}^{\infty}\alpha_k = \infty$; and 2) $\sum_{k=1}^{\infty}\alpha_k^2 < \infty$.*

Without a decreasing step-size, the estimation error from numerical approximations is asymptotically biased. One typical decaying step-size schedule is $\alpha_k = a(b + k)^{-\gamma}$, with $\gamma \in (0.5, 1]$ and $(a, b)$ some positive constants (Welling & Teh, 2011).

## 3 CYCLICAL SG-MCMC

We now introduce our *cyclical SG-MCMC* (cSG-MCMC) algorithm. cSG-MCMC consists of two stages: *exploration* and *sampling*. In the following, we first introduce the cyclical step-size schedule, and then describe the exploration stage in Section 3.1 and the sampling stage in Section 3.2. We propose an approach to combining samples for testing in Section F.

Assumption 1 guarantees the consistency of our estimation with the true distribution in the asymptotic time. The approximation error in limited time is characterized as the risk of an estimator $R = B^2 + V$, where $B$ is the bias and $V$ is the variance. In the case of infinite computation time, the traditional SG-MCMC setting can reduce the bias and variance to zero. However, the time budget is often limited in practice, and there is always a trade-off between bias and variance. We therefore decrease the overall approximation error $R$ by reducing the variance through obtaining more effective samples. The effective sample size can be increased if fewer correlated samples from different distribution modes are collected.

For deep neural networks, the parameter space is highly multimodal. In practice, SG-MCMC with the traditional decreasing stepsize schedule becomes trapped in a local mode, though injecting noise may help the sampler to escape in the asymptotic regime (Zhang et al., 2017). Inspired to improve the exploration of the multimodal posteriors for deep neural networks, with a simple and automatic approach, we propose the cyclical cosine stepsize schedule for SG-MCMC. The stepsize at iteration $k$ is defined as:

$$\alpha_k = \frac{\alpha_0}{2}\left[\cos\left(\frac{\pi \bmod(k - 1, \lceil K/M\rceil)}{\lceil K/M\rceil}\right) + 1\right], \tag{1}$$

where $\alpha_0$ is the initial stepsize, $M$ is the number of cycles and $K$ is the number of total iterations (Loshchilov & Hutter, 2016; Huang et al., 2017).

The stepsize $\alpha_k$ varies periodically with $k$. In each period, $\alpha_k$ starts at $\alpha_0$, and gradually decreases to 0. Within one period, SG-MCMC starts with a large stepsize, resulting in aggressive exploration in the parameter space; as the stepsize is decreasing, SG-MCMC explores local regions. In the next period, the Markov chain restarts with a large stepsize, encouraging the sampler to escape from the current mode and explore a new area of the posterior.

**Related work in optimization.** In optimization, the cyclical cosine annealing stepsize schedule has been demonstrated to be able to find diverse solutions in multimodal objectives, though not

specifically different modes, using stochastic gradient methods (Loshchilov & Hutter, 2016; Huang et al., 2017; Garipov et al., 2018; Fu et al., 2019). Alternatively, we adopt the technique to SG-MCMC as an effective scheme for sampling from multimodal distributions.

## 3.1 EXPLORATION

The first stage of cyclical SG-MCMC, *exploration*, discovers parameters near local modes of an objective function. Unfortunately, it is undesirable to directly apply the cyclical schedule in optimization to SG-MCMC for collecting samples at every step. SG-MCMC often requires a small stepsize in order to control the error induced by the noise from using a minibatch approximation. If the stepsize is too large, the stationary distribution of SG-MCMC might be far away from the true posterior distribution. To correct this error, it is possible to do stochastic Metropolis-Hastings (MH) (Korattikara et al., 2014; Bardenet et al., 2014; Chen et al., 2016b). However, stochastic MH correction is still computationally too expensive. Further, it is easy to get rejected with an aggressive large stepsize, and every rejection is a waste of gradient computations.

To alleviate this problem, we propose to introduce a system temperature $T$ to control the sampler's behaviour: $p(\theta|\mathcal{D}) \propto \exp(-U(\theta)/T)$. Note that the setting $T = 1$ corresponds to sampling from the untempered posterior. When $T \to 0$, the posterior distribution becomes a point mass. Sampling from $\lim_{T \to 0} \exp(-U(\theta)/T)$ is equivalent to minimizing $U(\theta)$; in this context, SG-MCMC methods become stochastic gradient optimization methods.

One may increase the temperature $T$ from 0 to 1 when the step-size is decreasing. We simply consider $T = 0$ and perform optimization as the burn-in stage, when the completed proportion of a cycle $r(k) = \frac{\mod (k-1, \lceil K/M \rceil)}{\lceil K/M \rceil}$ is smaller than a given threshold: $r(k) < \beta$. Note that $\beta \in (0, 1)$ balances the proportion of the exploration and sampling stages in cSG-MCMC.

## 3.2 SAMPLING

The *sampling* stage corresponds to $T = 1$ of the exploration stage. When $r(k) > \beta$ or step-sizes are sufficiently small, we initiate SG-MCMC updates and collect samples until this cycle ends.

**SG-MCMC with Warm Restarts.** One may consider the exploration stage as automatically providing warm restarts for the sampling stage. Exploration alleviates the inefficient mixing and inability to traverse the multimodal distributions of the traditional SG-MCMC methods. SG-MCMC with warm restarts explores different parts of the posterior distribution and captures multiple modes in a single training procedure.

---
**Algorithm 1** Cyclical SG-MCMC.

---
**Input:** The initial stepsize $\alpha_0$, number of cycles $M$, number of training iterations $K$ and the proportion of exploration stage $\beta$.
    **for** k = 1:K **do**
        $\alpha \leftarrow \alpha_k$ according to Eq equation 1.
        **if** $\frac{\mod (k-1, \lceil K/M \rceil)}{\lceil K/M \rceil} < \beta$ **then**
            % Exploration stage
            $\theta \leftarrow \theta - \alpha \nabla \tilde{U}_k(\theta)$
        **else**
            % Sampling stage
            Collect samples using SG-MCMC methods
**Output:** Samples $\{\theta_k\}$

---

In summary, the proposed cyclical SG-MCMC repeats the *two* stages, with three key advantages: (*i*) It restarts with a large stepsize at the beginning of a cycle which provides enough perturbation and encourages the model to escape from the current mode. (*ii*) The stepsize decreases more quickly inside one cycle than a traditional schedule, making the sampler better characterize the density of the local regions. (*iii*) This cyclical stepsize shares the advantage of the "super-convergence" property discussed in Smith & Topin (2017): cSG-MCMC can accelerate convergence for DNNs by up to an order of magnitude.

**Connection to the Santa algorithm.** It is interesting to note that our approach inverts steps of the Santa algorithm (Chen et al., 2016a) for optimization. Santa is a simulated-annealing-based optimization algorithm with an exploration stage when $T = 1$, then gradually anneals $T \to 0$ in a refinement stage for global optimization. In contrast, our goal is to draw samples for multimodal distributions, thus we explore with $T = 0$ and sample with $T = 1$. Another fundamental difference is that Santa adopts the traditional stepsize decay, while we use the cyclical schedule.

We visually compare the difference between cyclical and traditional step size schedules (described in Section 2) in Figure 1. The cyclical SG-MCMC algorithm is presented in Algorithm 1.

**Connection to Parallel MCMC.** Running parallel Markov chains is a natural and effective way to draw samples from multimodal distributions (VanDerwerken & Schmidler, 2013; Ahn et al., 2014). However, the training cost increases linearly with the number of chains. Cyclical SG-MCMC can be seen as an efficient way to approximate parallel MCMC. Each cycle effectively estimates a different region of posterior. Note cyclical SG-MCMC runs along a single training pass. Therefore, its computational cost is the same as single chain SG-MCMC while significantly less than parallel MCMC.

**Combining Samples.** In cyclical SG-MCMC, we obtain samples from multiple modes of a posterior distribution by running the cyclical step size schedule for many periods. We provide a sampling combination scheme to effectively use the collected samples in Section F in the appendix.

## 4 THEORETICAL ANALYSIS

Our algorithm is based on the SDE characterizing the Langevin dynamics: $\mathrm{d}\theta_t = -\nabla U(\theta_t)\mathrm{d}t + \sqrt{2}\mathrm{d}\mathcal{W}_t$, where $\mathcal{W}_t \in \mathbb{R}^d$ is a $d$-dimensional Brownian motion. In this section, we prove non-asymptotic convergence rates for the proposed cSG-MCMC framework with a cyclical stepsize sequence $\{\alpha_k\}$ defined in equation 1. For simplicity, we do not consider the exploration stage in the analysis as that corresponds to stochastic optimization. Generally, there are two different ways to describe the convergence behaviours of SG-MCMC. One characterizes the sample average over a particular test function (*e.g.*, Chen et al. (2015); Vollmer et al. (2016)); the other is in terms of the Wasserstein distance (*e.g.*, Raginsky et al. (2017); Xu et al. (2017)). We study both in the following.

**Weak convergence** Following Chen et al. (2015) and Vollmer et al. (2016), we define the posterior average of an ergodic SDE as: $\bar{\phi} \triangleq \int_{\mathcal{X}} \phi(\theta)\rho(\theta)\mathrm{d}\theta$ for some test function $\phi(\theta)$ of interest. For the corresponding algorithm with generated samples $(\theta_k)_{k=1}^K$, we use the *sample average* $\hat{\phi}$ defined as $\hat{\phi} = \frac{1}{K}\sum_{k=1}^K \phi(\theta_k)$ to approximate $\bar{\phi}$. We prove weak convergence of cSGLD in terms of bias and MSE, as stated in Theorem 1.

**Theorem 1.** *Under Assumptions 2 in the appendix, for a smooth test function $\phi$, the bias and MSE of cSGLD are bounded as:*

$$\text{BIAS: } \left|\mathbb{E}\tilde{\phi} - \bar{\phi}\right| = O\left(\frac{1}{\alpha_0 K} + \alpha_0\right), \quad \text{MSE: } \mathbb{E}\left(\tilde{\phi} - \bar{\phi}\right)^2 = O\left(\frac{1}{\alpha_0 K} + \alpha_0^2\right). \quad (2)$$

**Convergence under the Wasserstein distance** Next, we consider the more general case of SGLD and characterize convergence rates in terms of a stronger metric of 2-Wasserstein distance, defined as:

$$W_2^2(\mu, \nu) := \inf_\gamma \left\{ \int_{\Omega \times \Omega} \|\theta - \theta'\|_2^2 \mathrm{d}\gamma(\theta, \theta') : \gamma \in \Gamma(\mu, \nu) \right\}$$

where $\Gamma(\mu, \nu)$ is the set of joint distributions over $(\theta, \theta')$ such that the two marginals equal $\mu$ and $\nu$, respectively.

Denote the distribution of $\theta_t$ in the SDE as $\nu_t$. According to Chiang & Hwang (1987), the stationary distribution $\nu_\infty$ matches our target distribution. Let $\mu_K$ be the distribution of the sample from our proposed cSGLD algorithm at the $K$-th iteration. Our goal is to derive a convergence bound on $W_2(\mu_K, \nu_\infty)$. We adopt standard assumptions as in most existing work, which are detailed in Assumption 3 in the appendix. Theorem 2 summarizes our main theoretical result.

**Theorem 2.** *Under Assumption 3 in the appendix, there exist constants $(C_0, C_1, C_2, C_3)$ independent of the stepsizes such that the convergence rate of our proposed cSGLD with cyclical stepsize sequence equation 1 is bounded for all $K$ satisfying ($K \mod M = 0$), as $W_2(\mu_K, \nu_\infty) \leq$*

$$C_3 \exp(-\frac{K\alpha_0}{2C_4}) + \left(6 + \frac{C_2 K\alpha_0}{2}\right)^{\frac{1}{2}} [(C_1 \frac{3\alpha_0^2 K}{8} + \sigma C_0 \frac{K\alpha_0}{2})^{\frac{1}{2}} + (C_1 \frac{3\alpha_0^2 K}{16} + \sigma C_0 \frac{K\alpha_0}{4})^{\frac{1}{4}}].$$

*Particularly, if we further assume $\alpha_0 = O(K^{-\beta})$ for $\forall \beta > 1$, $W_2(\mu_K, \nu_\infty) \leq C_3 + \left(6 + \frac{C_2}{K^{\beta-1}}\right)^{\frac{1}{2}} [(\frac{2C_1}{K^{2\beta-1}} + \frac{2C_0}{K^{\beta-1}})^{\frac{1}{2}} + (\frac{C_1}{K^{2\beta-1}} + \frac{C_0}{K^{\beta-1}})^{\frac{1}{4}}].$*

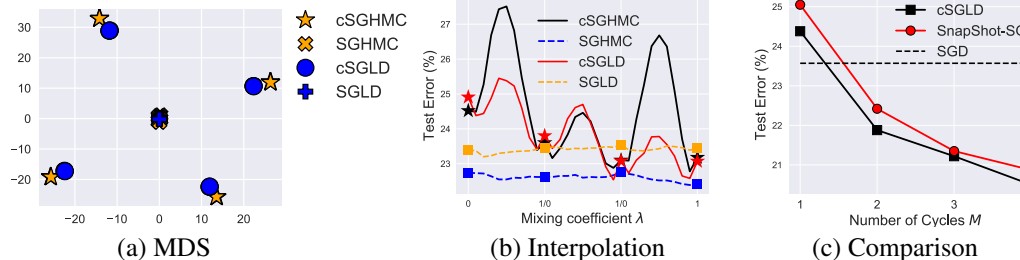

| (a) MDS | (b) Interpolation | (c) Comparison |

Figure 3: Results of cSG-MCMC with DNNs on the CIFAR-100 dataset. (a) MDS visualization in weight space: cSG-MCMC show larger distance than traditional schedules. (b) Testing errors (%) on the path of two samples: cSG-MCMC shows more varied performance. (c) Testing errors (%) as a function of the number of cycles $M$: cSGLD yields consistently lower errors.

**Remark 1.** *i) The bound is decomposed into two parts: the first part measures convergence speed of exact solution to the stationary distribution, i.e., $\nu_{\sum_k \alpha_k}$ to $\nu_\infty$; the second part measures the numerical error, i.e., between $\mu_K$ and $\nu_{\sum_k \alpha_k}$. ii) The overall bound offers a same order of dependency on $K$ as in standard SGLD (please see the bound for SGLD in Section E of the appendix. See also Raginsky et al. (2017)). iii) If one imposes stricter assumptions such as in the convex case, the bound can be further improved. Specific bounds are derived in the appendix. We did not consider this case due to the discrepancy from real applications.*

## 5 EXPERIMENTS

We demonstrate cSG-MCMC on several tasks, including a synthetic multimodal distribution (Section 5.1), image classification on Bayesian neural networks (Section 5.2) and uncertainty estimation in Section 5.3. We also demonstrate cSG-MCMC can improve the estimate efficiency for uni-modal distributions using Bayesian logistic regression in Section A.2 in the appendix. We choose SLGD and SGHMC as the representative baseline algorithms. Their cyclical counterpart are called cSGLD and cSGHMC, respectively.

### 5.1 SYNTHETIC MULTIMODAL DATA

We first demonstrate the ability of cSG-MCMC for sampling from a multi-modal distribution on a 2D mixture of 25 Gaussians. Specifically, we compare cSGLD with SGLD in two setting: (1) parallel running with 4 chains and (2) running with a single chain, respectively. Each chain runs for 50k iterations. The step-size schedule of SGLD is $\alpha_k \propto 0.05k^{-0.55}$. In cSGLD, we set $M = 30$ and the initial step-size $\alpha_0 = 0.09$. The proportion of exploration stage $\beta = \frac{1}{4}$. Fig 2 shows the estimated density using sampling results for SGLD and cSGLD in the parallel setting. We observed that SGLD gets trapped in the local modes, depending on the initial position. In any practical time period,

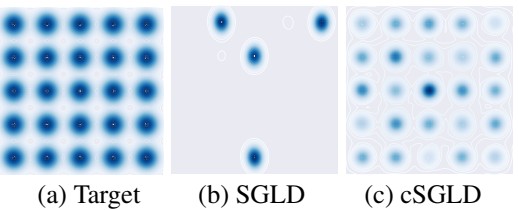

| (a) Target | (b) SGLD | (c) cSGLD |

Figure 2: Sampling from a mixture of 25 Gaussians shown in (a) for the parallel setting. With a budget of $50\text{k} \times 4 = 200\text{k}$ samples, traditional SGLD in (b) has only discovered 4 of the 25 modes, while our cSGLD in (c) has fully explored the distribution.

SGLD could only characterize partial distribution. In contrast, cSGLD is able to find and characterize all modes, regardless of the initial position. cSGLD leverages large step sizes to discover a new mode, and small step sizes to explore local modes. This result suggests cSGLD can be a significantly favourable choice in the non-asymptotic setting, for example only 50k iterations in this case. The single chain results and the quantitative results on mode coverage are reported in Section A.1 of the appendix.

### 5.2 BAYESIAN NEURAL NETWORKS

We demonstrate the effectiveness of cSG-MCMC on Bayesian neural networks for classification on CIFAR-10 and CIFAR-100. We compare with (*i*) traditional SG-MCMC; (*ii*) traditional stochastic optimization methods, including stochastic gradient descent (SGD) and stochastic gradient descent

with momentum (SGDM); and (*iii*) *Snapshot*: a stochastic optimization ensemble method method with a the cyclical stepsize schedule (Huang et al., 2017). We use a ResNet-18 (He et al., 2016) and run all algorithms for 200 epochs. We report the test errors averaged over 3 runs, and the standard error ($\pm$) from the mean predictor.

We set $M = 4$ and $\alpha_0 = 0.5$ for cSGLD, cSGHMC and Snapshot. The proportion hyper-parameter $\beta = 0.8$ and 0.94 for CIFAR-10 and CIFAR-100, respectively. We collect 3 samples per cycle. In practice, we found that the collected samples share similarly high likelihood for DNNs, thus one may simply set the normalizing term $w_i$ in equation 33 to be the same for faster testing.

We found that *tempering* helps improve performance for Bayesian inference with neural networks. Tempering for SG-MCMC was first used by Li et al. (2016a) as a practical technique for neural network training for fast convergence in limited time[1]. We simply use the prescribed temperature of Li et al. (2016a) without tuning, but better results of the sampling methods can be achieved by tuning the temperature. More details are in Appendix J. We hypothesize that tempering helps due to the overparametrization of neural networks. Tempering enables one to leverage the inductive biases of the network, while representing the belief that the model capacity can be misspecified. In work on *Safe Bayes*, also known as *generalized* and *fractional* Bayesian inference, tempered posteriors are well-known to help under misspecification (e.g., Barron & Cover, 1991; de Heide et al., 2019; Grünwald et al., 2017).

For the traditional SG-MCMC methods, we found that noise injection early in training hurts convergence. To make these baselines as competitive as possible, we thus avoid noise injection for the first 150 epochs of training (corresponding to the zero temperature limit of SGLD and SGHMC), and resume SGMCMC as usual (with noise) for the last 50 epochs. This scheme is similar to the exploration and sampling stages within one cycle of cSG-MCMC. We collect 20 samples for the MCMC methods and average their predictions in testing.

**Testing Performance for Image Classification**
We report the testing errors in Table 1 to compare with the non-parallel algorithms. Snapshot and traditional SG-MCMC reduce the testing errors on both datasets. Performance variance for these methods is also relatively small, due to the multiple networks in the Bayesian model average. Further, cSG-MCMC significantly outperforms Snapshot ensembles and the traditional SG-MCMC, demonstrating the importance of (1) capturing diverse modes compared to traditional SG-MCMC, and (2) capturing fine-scale characteristics of the distribution compared with Snapshot ensembles.

|  | CIFAR-10 | CIFAR-100 |
|---|---|---|
| SGD | 5.29$\pm$0.15 | 23.61$\pm$0.09 |
| SGDM | 5.17$\pm$0.09 | 22.98$\pm$0.27 |
| Snapshot-SGD | 4.46$\pm$0.04 | 20.83$\pm$0.01 |
| Snapshot-SGDM | 4.39$\pm$0.01 | 20.81$\pm$0.10 |
| SGLD | 5.20$\pm$0.06 | 23.23$\pm$0.01 |
| cSGLD | 4.29$\pm$0.06 | 20.55$\pm$0.06 |
| SGHMC | 4.93$\pm$0.1 | 22.60$\pm$0.17 |
| cSGHMC | **4.27**$\pm$0.03 | **20.50**$\pm$0.11 |

Table 1: Comparison of test error (%) between cSG-MCMC with non-parallel algorithms. cSGLD and cSGHMC yields lower errors than their optimization counterparts, respectively.

**Diversity in Weight Space.** To further demonstrate our hypothesis that with a limited budget cSG-MCMC can find diverse modes, while traditional SG-MCMC cannot, we visualize the 12 samples we collect from cSG-MCMC and SG-MCMC on CIFAR-100 respectively using Multidimensional Scaling (MDS) in Figure 3 (a). MDS uses a Euclidean distance metric between the weight of samples. We see that the samples of cSG-MCMC form 4 clusters, which means they are from 4 different modes in weight space. However, all samples from SG-MCMC only form one cluster, which indicates traditional SG-MCMC gets trapped in one mode and only samples from that mode.

**Diversity in Prediction.** To further demonstrate the samples from different cycles of cSG-MCMC provide diverse predictions we choose one sample from each cycle and linearly interpolate between two of them (Goodfellow et al., 2014; Huang et al., 2017). Specifically, let $J(\theta)$ be the test error of a sample with parameter $\theta$. We compute the test error of the convex combination of two samples $J(\lambda\theta_1 + (1 - \lambda)\theta_2)$, where $\lambda \in [0, 1]$.

We linearly interpolate between two samples from neighboring chains of cSG-MCMC since they are the most likely to be similar. We randomly select 4 samples from SG-MCMC. If the samples are from the same mode, the test error of the linear interpolation of parameters will be relatively

---

[1]https://github.com/ChunyuanLI/pSGLD/issues/2

| Method | Cyclical+Parallel | | Decreasing+Parallel | | Decreasing+Parallel | | Cyclical+Single | |
|---|---|---|---|---|---|---|---|---|
| Cost | 200/800 | | 200/800 | | 100/400 | | 200/200 | |
| Sampler | SGLD | SGHMC | SGLD | SGHMC | SGLD | SGHMC | SGLD | SGHMC |
| CIFAR-10 | 4.09 | 3.95 | 4.15 | 4.09 | 5.11 | 4.52 | 4.29 | 4.27 |
| CIFAR-100 | 19.37 | 19.19 | 20.29 | 19.72 | 21.16 | 20.82 | 20.55 | 20.50 |

Table 2: Comparison of test error (%) between cSG-MCMC with parallel algorithm ($M$=4 chains) on CIFAR-10 and CIFAR-100. The method is reported in the format of "step-size schedule (cyclical or decreasing) + single/parallel chain". The cost is reported in the format of "#epoch per chain / #epoch used in all chains". Note that a parallel algorithm with a single chain reduces to a non-parallel algorithm. Integration of the cyclical schedule with parallel algorithms provides lower testing errors.

smooth, while if the samples are from different modes, the test error of the parameter interpolation will have a spike when $\lambda$ is between 0 and 1.

We show the results of interpolation for cSG-MCMC and SG-MCMC on CIFAR-100 in Figure 3 (b). We see a spike in the test error in each linear interpolation of parameters between two samples from neighboring chains in cSG-MCMC while the linear interpolation for samples of SG-MCMC is smooth. This result suggests that samples of cSG-MCMC from different chains are from different modes while samples of SG-MCMC are from the same mode.

Although the test error of a single sample of cSG-MCMC is worse than that of SG-MCMC shown in Figure 3 (c), the ensemble of these samples significantly improves the test error, indicating that samples from different modes provide different predictions and make mistakes on different data points. Thus these diverse samples can complement each other, resulting in a lower test error, and demonstrating the advantage of exploring diverse modes using cSG-MCMC.

**Comparison to Parallel MCMC.** cSG-MCMC can be viewed as an economical alternative to parallel MCMC. We verify how closely cSG-MCMC can approximate the performance of parallel MCMC, but with more convenience and less computational expense. We also note that we can improve parallel MCMC with the proposed cyclical stepsize schedule.

We report the testing errors in Table 2 to compare multiple-chain results. (1) Four chains used, each runs 200 epochs (800 epochs in total), the results are shown in the first 4 columns (Cyclical+Parallel vs Decreasing+Parallel). We see that cSG-MCMC variants provide lower errors than plain SG-MCMC. (2) We reduce the number of epochs (#epoch) of parallel MCMC to 100 epoch each for decreasing stepsize schedule. The total cost is 400 epochs. We compare its performance with cyclical single chain (200 epochs in total) in the last 4 columns (Decreasing+Parallel vs Cyclical+Single). We see that the cyclical schedule running on a single chain performs best even with half the computational cost! All the results indicate the importance of warm re-starts using the proposed cyclical schedule. For a given total cost budget, the proposed cSGMCMC is preferable to parallel sampling.

**Comparison to Snapshot Optimization.** We carefully compared with Snapshot, as our cSG-MCMC can be viewed as the sampling counterpart of the Snapshot optimization method. We plot the test error *wrt.* various number of cycles $M$ in Fig. 3. As $M$ increases, cSG-MCMC and Snapshot both improve. However, given a fixed $M$, cSG-MCMC yields substantially lower test errors than Snapshot. This result is due to the ability of cSG-MCMC to better characterize the local distribution of modes: Snapshot provides a singe minimum per cycle, while cSG-MCMC fully exploits the mode with more samples, which could provide weight uncertainty estimate and avoid over-fitting.

**Results on ImageNet.** We further study different learning algorithms on a large-scale dataset, ImageNet. ResNet-50 is used as the architecture, and 120 epochs for each run. The results on the testing set are summarized in Table 3, including NLL, Top1 and Top5 accuracy (%), respectively. 3 cycles are considered for both cSGHMC and Snapshot, and we collect 3 samples per cycle. We see that cSGHMC yields the lowest testing NLL, indicating that the cy-

| | NLL ↓ | Top1 ↑ | Top5 ↑ |
|---|---|---|---|
| SGDM | 0.9595 | 76.046 | 92.776 |
| Snapshot-SGDM | 0.8941 | 77.142 | 93.344 |
| SGHMC | 0.9308 | 76.274 | 92.994 |
| cSGHMC | **0.8882** | 77.114 | 93.524 |

Table 3: Comparison on the testing set of ImageNet. cSGHMC yields lower testing NLL than Snapshot and SGHMC.

cle schedule is an effective technique to explore the parameter space, and diversified samples can help prevent over-fitting.

### 5.3 UNCERTAINTY EVALUATION

To demonstrate how predictive uncertainty benefits from exploring multiple modes in the posterior of neural network weights, we consider the task of uncertainty estimation for out-of-distribution samples (Lakshminarayanan et al., 2017). We train a three-layer MLP model on the standard MNIST train dataset until convergence using different algorithms, and estimate the entropy of the predictive distribution on the notMNIST dataset (Bulatov, 2011). Since the samples from the notMNIST dataset belong to the unseen classes, ideally the predictive distribution of the trained model should be uniform over the notMNIST digits, which gives the maximum entropy.

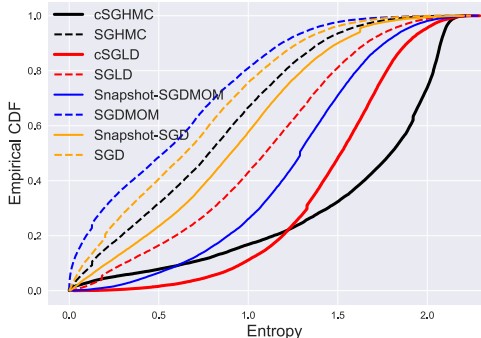

Figure 4: Empirical CDF for the entropy of the predictive distribution on notMNIST dataset. cSGLD and cSGHMC show lower probability for the low entropy estimate than other algorithms.

In Figure 4, we plot the empirical CDF for the entropy of the predictive distributions on notMNIST. We see that the uncertainty estimates from cSGHMC and cSGLD are better than the other methods, since the probability of a low entropy prediction is overall lower. cSG-MCMC algorithms explore more modes in the weight space, each mode characterizes a meaningfully different representation of MNIST data. When testing on the out-of-distribution dataset (notMNIST), each mode can provide different predictions over the label space, leading to more reasonable uncertainty estimates. Snapshot achieves less entropy than cSG-MCMC, since it represents each mode with a single point.

The traditional SG-MCMC methods also provide better uncertainty estimation compared to their optimization counterparts, because they characterize a local region of the parameter space, rather than a single point. cSG-MCMC can be regarded as a combination of these two worlds: a wide coverage of many modes in Snapshot, and fine-scale characterization of local regions in SG-MCMC.

## 6 DISCUSSION

We have proposed cyclical SG-MCMC methods to automatically explore complex multimodal distributions. Our approach is particularly compelling for Bayesian deep learning, which involves rich multimodal parameter posteriors corresponding to meaningfully different representations. We have also shown that our cyclical methods explore unimodal distributions more efficiently. These results are in accordance with theory we developed to show that cyclical SG-MCMC will converge faster to samples from a stationary distribution in general settings. Moreover, we show cyclical SG-MCMC methods provide more accurate uncertainty estimation, by capturing more diversity in the hypothesis space corresponding to settings of model parameters.

While MCMC was once the gold standard for inference with neural networks, it is now rarely used in modern deep learning. We hope that this paper will help renew interest in MCMC for posterior inference in deep learning. Indeed, MCMC is uniquely positioned to explore the rich multimodal posterior distributions of modern neural networks, which can lead to improved accuracy, reliability, and uncertainty representation.

### ACKNOWLEDGEMENTS

AGW was supported by an Amazon Research Award, Facebook Research, NSF I-DISRE 193471, NIH R01 DA048764-01A1, NSF IIS-1563887, and NSF IIS-1910266.

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

## A EXPERIMENTAL RESULTS

### A.1 SYNTHETIC MULTIMODAL DISTRIBUTION

The density of the distribution is

$$F(x) = \sum_{i=1}^{25} \lambda \mathcal{N}(x|\mu_i, \Sigma),$$

where $\lambda = \frac{1}{25}, \mu = \{-4, -2, 0, 2, 4\}^\top \times \{-4, -2, 0, 2, 4\}, \Sigma = \begin{bmatrix} 0.03 & 0 \\ 0 & 0.03 \end{bmatrix}$.

In Figure 5, we show the estimated density for SGLD and cSGLD in the non-parallel setting.

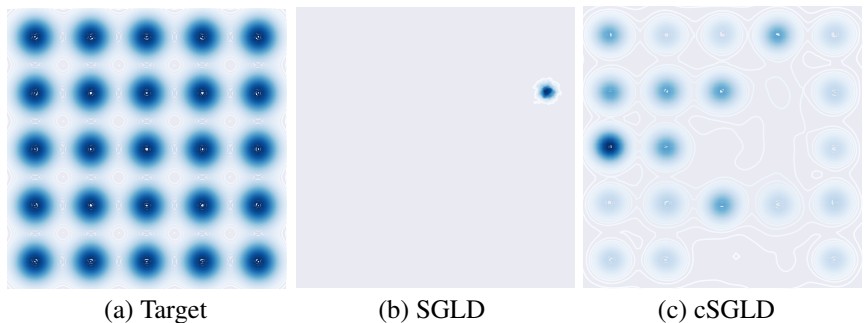

|           (a) Target           |           (b) SGLD           |           (c) cSGLD           |

Figure 5: Sampling from a mixture of 25 Gaussians in the non-parallel setting. With a budget of 50K samples, traditional SGLD has only discovered one of the 25 modes, while our proposed cSGLD has explored significantly more of the distribution.

To quantitatively show the ability of different algorithms to explore multi-modal distributions, we define the *mode-coverage* metric: when the number of samples falling within the radius $r$ of a mode center is larger than a threshold $\bar{n}$, we consider this mode covered. On this dataset, we choose $r = 0.25$ and $\bar{n} = 100$. Table 4 shows the mode-coverage for several algorithms, based on 10 different runs.

| Algorithm | Mode coverage |
|---|---|
| SGLD | 1.8±0.13 |
| cSGLD | **6.7**±0.52 |
| Parallel SGLD | 18±0.47 |
| Parallel cSGLD | **24.4**±0.22 |

Table 4: Mode coverage over 10 different runs, ± standard error.

### A.2 BAYESIAN LOGISTIC REGRESSION

We consider Bayesian logistic regression (BLR) on three real-world datasets from the UCI repository: *Australian* (15 covariates, 690 data points), *German* (25 covariates, 1000 data points) and *Heart* (14 covariates, 270 data points). For all experiments, we collect 5000 samples with 5000 burn-in iterations. Following the settings in Li et al. (2016a), we report median *effective sample size* (ESS) in Table 5.

Note that BLR is *unimodal* in parameter space. We use this experiment as an adversarial situation for cSG-MCMC, which we primarily designed to explore multiple modes. We note that even in the unimodal setting, cSG-MCMC more effectively explores the parameter space than popular alternatives. We can also use these experiments to understand how samplers respond to varying parameter dimensionality and training set sizes.

Overall, cSG-MCMC dramatically outperforms SG-MCMC, which demonstrates the fast mixing rate due to the warm restarts. On the small dataset *Heart*, SGHMC and cSGHMC achieve the

same results, because the posterior of BLR on this dataset is simple. However, in higher dimensional spaces (*e.g., Australian* and *German*), cSG-MCMC shows significantly higher ESS; this result means that each cycle in cSG-MCMC can characterize a different region of the posteriors, combining multiple cycles yields more accurate overall approximation.

|       | Australian | German | Heart |
|-------|------------|--------|-------|
| SGLD  | 1676       | 492    | 2199  |
| cSGLD | 2138       | 978    | 2541  |
| SGHMC | 1317       | 2007   | **5000** |
| cSGHMC| **4707**   | **2436** | **5000** |

Table 5: Effective sample size for samples for the unimodal posteriors in Bayesian linear regression, obtained using cyclical and traditional SG-MCMC algorithms, respectively.

# B    ASSUMPTIONS

## B.1    ASSUMPTIONS IN WEAK CONVERGENCE ANALYSIS

In the analysis, we define a functional $\psi$ that solves the following *Poisson Equation*:

$$\mathcal{L}\psi(\theta_k) = \phi(\theta_k) - \bar{\phi}, \text{ or equivalently, } \frac{1}{K}\sum_{k=1}^{K}\mathcal{L}\psi(\theta_k) = \hat{\phi} - \bar{\phi}. \tag{3}$$

The solution functional $\psi(\theta_k)$ characterizes the difference between $\phi(\theta_k)$ and the posterior average $\bar{\phi}$ for every $\theta_k$, thus would typically possess a unique solution, which is at least as smooth as $\phi$ under the elliptic or hypoelliptic settings (Mattingly et al., 2010). Following Chen et al. (2015); Vollmer et al. (2016), we make certain assumptions on the solution functional, $\psi$, of the Poisson equation equation 3.

**Assumption 2.** *$\psi$ and its up to 3rd-order derivatives, $\mathcal{D}^k\psi$, are bounded by a function $\mathcal{V}$, i.e., $\|\mathcal{D}^k\psi\| \leq H_k\mathcal{V}^{p_k}$ for $k = (0,1,2,3)$, $H_k, p_k > 0$. Furthermore, the expectation of $\mathcal{V}$ on $\{\theta_k\}$ is bounded: $\sup_l \mathbb{E}\mathcal{V}^p(\theta_k) < \infty$, and $\mathcal{V}$ is smooth such that $\sup_{s\in(0,1)}\mathcal{V}^p(s\theta + (1-s)\theta') \leq C(\mathcal{V}^p(\theta) + \mathcal{V}^p(\theta'))$, $\forall \theta, \theta', p \leq \max\{2p_k\}$ for some $C > 0$.*

## B.2    ASSUMPTIONS IN CONVERGENCE UNDER THE WASSERSTEIN DISTANCE

Following existing work in Raginsky et al. (2017), we adopt the following standard assumptions summarized in Assumption 3.

**Assumption 3.**
- *There exists some constants $A \geq 0$ and $B \geq 0$, such that $U(0) \leq A$ and $\nabla U(0) \leq B$.*

- *The function $U$ is $L_U$-smooth : $\|\nabla U(w) - \nabla U(v)\| \leq L_U\|w - v\|$.*

- *The function $U$ is $(m_u, b) - dissipative$, which means for some $m_U > 0$ and $b > 0$ $\langle w, \nabla U(w)\rangle \geq m_U\|w\|^2 - b$.*

- *There exists some constant $\delta \in [0,1)$, such that $\mathbb{E}[\|\nabla\tilde{U}_k(w) - \nabla U(w)\|^2] \leq 2\sigma(M_U^2\|w\|^2 + B^2)$.*

- *We can choose $\mu_0$ which satisfies the requirement: $\kappa_0 := \log\int e^{\|w\|^2}\mu_0(w)dw < \infty$.*

# C    PROOF OF THEOREM 1

To prove the theorem, we borrow tools developed by Chen et al. (2015); Vollmer et al. (2015). We first rephrase the stepsize assumptions in general SG-MCMC in Assumption 4.

**Assumption 4.** *The algorithm adopts an N-th order integrator. The step sizes $\{h_k\}$ are such that $0 < h_{k+1} < h_k$, and satisfy 1) $\sum_{k=1}^{\infty}h_k = \infty$; and 2) $\lim_{K\to\infty}\frac{\sum_{k=1}^{K}h_k^{N+1}}{\sum_{k=1}^{K}h_k} = 0$.*

Our prove can be derived by the following results from Chen et al. (2015).

**Lemma 1** (Chen et al. (2015)). *Let $S_K \triangleq \sum_{k=1}^{K} h_k$. Under Assumptions 2 and 4, for a smooth test function $\phi$, the bias and MSE of a decreasing-step-size SG-MCMC with a $N$th-order integrator at time $S_L$ are bounded as:*

$$\text{BIAS: } \left| \mathbb{E}\tilde{\phi} - \bar{\phi} \right| = O\left( \frac{1}{S_K} + \frac{\sum_{k=1}^{K} h_k^{N+1}}{S_K} \right) \tag{4}$$

$$\text{MSE: } \mathbb{E}\left( \tilde{\phi} - \bar{\phi} \right)^2 \leq C\left( \sum_l \frac{h_k^2}{S_K^2} \mathbb{E}\left\| \Delta V_l \right\|^2 + \frac{1}{S_K} + \frac{(\sum_{k=1}^{K} h_k^{N+1})^2}{S_K^2} \right). \tag{5}$$

Note that Assumption 4 is only required if one wants to prove the asymptotically unbias of an algorithm. Lemma 1 still applies even if Assumption 4 is not satisfied. In this case one would obtain a biased algorithm, which is the case of cSGLD.

*Proof of Theorem 1.* Our results is actually a special case of Lemma 1. To see that, first note that our cSGLD adopts a first order integrator, thus $N = 1$. To proceed, note that $S_K = \sum_{k=1}^{K} \alpha_k = O(\alpha_0 K)$, and

$$\sum_{j=0}^{K-1} \alpha_{j+1}^2 = \frac{\alpha_0^2}{4} \sum_{j=0}^{K-1} [\cos(\frac{\pi mod(j-1, [K/M])}{[K/M]}) + 1]^2$$

$$= \frac{\alpha_0^2}{4} \sum_{j=0}^{K-1} [\cos^2(\frac{\pi mod(j-1, K/M)}{K/M}) + 1]^2$$

$$= \frac{\alpha_0^2}{4} \frac{K}{M}(\frac{M}{2} + M) = \frac{3\alpha_0^2 K}{8}. \tag{6}$$

As a result, for the bias, we have

$$\left| \mathbb{E}\tilde{\phi} - \bar{\phi} \right| = O\left( \frac{1}{S_K} + \frac{\sum_{k=1}^{K} h_k^{N+1}}{S_K} \right) = O\left( \frac{1}{\alpha_0 K} + \frac{3\alpha_0^2 K/8}{\alpha_0 K} \right)$$

$$= O\left( \frac{1}{\alpha_0 K} + \alpha_0 \right).$$

For the MSE, note the first term $\sum_l \frac{h_k^2}{S_K^2} \mathbb{E}\left\| \Delta V_l \right\|^2$ has a higher order than other terms, thus it is omitted in the big-O notation, *i.e.*,

$$\mathbb{E}\left( \tilde{\phi} - \bar{\phi} \right)^2 = O\left( \frac{1}{\alpha_0 K} + (\frac{3\alpha_0^2 K/8}{\alpha_0 K})^2 \right)$$

$$= O\left( \frac{1}{\alpha_0 K} + \alpha_0^2 \right).$$

This completes the proof. □

## D  PROOF OF THEOREM 2

*Proof of the bound for $W_2(\mu_K, \nu_\infty)$ in cSGLD.* Firstly, we introduce the following SDE

$$d\theta_t = -\nabla U(\theta_t)dt + \sqrt{2}d\mathcal{W}_t, \tag{7}$$

Let $\nu_t$ denote the distribution of $\theta_t$, and the stationary distribution of equation 34 be $p(\theta|\mathcal{D})$, which means $\nu_\infty = p(\theta|\mathcal{D})$.

$$\theta_{k+1} = \theta_k - \nabla \tilde{U}_k(\theta_k)\alpha_{k+1} + \sqrt{2\alpha_{k+1}}\xi_{k+1} \tag{8}$$

Further, let $\mu_k$ denote the distribution of $\theta_k$.

Since

$$W_2(\mu_K, \nu_\infty) \leq W_2(\mu_K, \nu_{\sum_{k=1}^{K} \alpha_k}) + W_2(\nu_{\sum_{k=1}^{K} \alpha_k}, \nu_\infty) \tag{9}$$

, we need to give the bounds for these two parts respectively.

## D.1  $W_2(\mu_K, \nu_{\sum_{k=1}^K \alpha_k})$

For the first part, $W_2(\mu_K, \nu_{\sum_{k=1}^K \alpha_k})$, our proof is based on the proof of Lemma 3.6 in Raginsky et al. (2017) with some modifications. We first assume $\mathbb{E}(\nabla \tilde{U}(w)) = \nabla U(w), \ \forall w \in \mathbb{R}^d$, which is a general assumption according to the way we choose the minibatch. And we define $p(t)$ which will be used in the following proof:

$$p(t) = \{k \in \mathbb{Z} | \sum_{i=1}^{k} \alpha_i \leq t < \sum_{i=1}^{k+1} \alpha_i\} \tag{10}$$

Then we focus on the following continuous-time interpolation of $\theta_k$:

$$\underline{\theta}(t) = \theta_0 - \int_0^t \nabla \tilde{U} \left( \underline{\theta}(\sum_{k=1}^{p(s)} \alpha_k) \right) ds + \sqrt{2} \int_0^t d\mathcal{W}_s^{(d)} \tag{11}$$

where $\nabla \tilde{U} \equiv \nabla \tilde{U}_k$ for $t \in \left[ \sum_{i=1}^{k} \alpha_i, \sum_{i=1}^{k+1} \alpha_i \right)$. And for each k , $\underline{\theta}(\sum_{i=1}^{k} \alpha_i)$ and $\theta_k$ have the same probability law $\mu_k$.
Since $\underline{\theta}(t)$ is not a Markov process, we define the following process which has the same one-time marginals as $\underline{\theta}(t)$

$$V(t) = \theta_0 - \int_0^t G_s(V(s)) \, ds + \sqrt{2} \int_0^t d\mathcal{W}_s^{(d)} \tag{12}$$

with

$$G_t(x) := \mathbb{E} \left[ \nabla \tilde{U} \left( \underline{\theta}(\sum_{i=1}^{q(t)} \alpha_i) \right) | \underline{\theta}(t) = x \right] \tag{13}$$

Let $\mathbf{P}_V^t := \mathcal{L}(V(s) : 0 \leq s \leq t)$ and $\mathbf{P}_\theta^t := \mathcal{L}(\theta(s) : 0 \leq s \leq t)$ and according to the proof of Lemma 3.6 in Raginsky et al. (2017), we can derive a similar result for the relative entropy of $\mathbf{P}_V^t$ and $\mathbf{P}_\theta^t$:

$$
\begin{aligned}
D_{KL}(\mathbf{P}_V^t \| \mathbf{P}_\theta^t) &= - \int d\mathbf{P}_V^t \log \frac{d\mathbf{P}_V^t}{d\mathbf{P}_\theta^t} \\
&= \frac{1}{4} \int_0^t \mathbb{E}\|\nabla U(V(s)) - G_s(V(s))\|^2 ds \\
&= \frac{1}{4} \int_0^t \mathbb{E}\|\nabla U(\underline{\theta}(s)) - G_s(\underline{\theta}(s))\|^2 ds
\end{aligned}
$$

The last line follows the fact that $\mathcal{L}(\underline{\theta}(s)) = \mathcal{L}(V(s))$, $\forall s$.

Then we will let $t = \sum_{k=1}^{K} \alpha_k$ and we can use the martingale property of the integral to derive:

$$D_{KL}(\mathbf{P}_V^{\sum_{k=1}^{K} \alpha_k} \| \mathbf{P}_{\theta}^{\sum_{k=1}^{K} \alpha_k})$$

$$= \frac{1}{4} \sum_{j=0}^{K-1} \int_{\sum_{k=1}^{j} \alpha_k}^{\sum_{k=1}^{j+1} \alpha_k} \mathbb{E}\|\nabla U(\underline{\theta}(s)) - G_s(\underline{\theta}(s))\|^2 \mathrm{d}s$$

$$\leq \frac{1}{2} \sum_{j=0}^{K-1} \int_{\sum_{k=1}^{j} \alpha_k}^{\sum_{k=1}^{j+1} \alpha_k} \mathbb{E}\|\nabla U(\underline{\theta}(s)) - \nabla U(\underline{\theta}(\sum_{k=1}^{q(s)} \alpha_i))\|^2 \mathrm{d}s$$

$$+ \frac{1}{2} \sum_{j=0}^{K-1} \int_{\sum_{k=1}^{j} \alpha_k}^{\sum_{k=1}^{j+1} \alpha_k} \mathbb{E}\|\nabla U(\underline{\theta}(\sum_{k=1}^{q(s)} \alpha_i)) - G_s(\underline{\theta}(\sum_{k=1}^{q(s)} \alpha_i))\|^2 \mathrm{d}s$$

$$\leq \frac{L_U^2}{2} \sum_{j=0}^{K-1} \int_{\sum_{k=1}^{j} \alpha_k}^{\sum_{k=1}^{j+1} \alpha_k} \mathbb{E}\|\underline{\theta}(s) - \underline{\theta}(\sum_{k=1}^{q(s)} \alpha_i)\|^2 \mathrm{d}s \tag{14}$$

$$+ \frac{1}{2} \sum_{j=0}^{K-1} \int_{\sum_{k=1}^{j} \alpha_k}^{\sum_{k=1}^{j+1} \alpha_k} \mathbb{E}\|\nabla U(\underline{\theta}(\sum_{k=1}^{q(s)} \alpha_i)) - G_s(\underline{\theta}(\sum_{k=1}^{q(s)} \alpha_i))\|^2 \mathrm{d}s \tag{15}$$

For the first part (14), we consider some $s \in [\sum_{k=1}^{j} \alpha_k, \sum_{k=1}^{j+1} \alpha_k)$, for which the following holds:

$$\underline{\theta}(s) - \underline{\theta}(\sum_{k=1}^{j} \alpha_k)$$

$$= -(s - \sum_{k=1}^{j} \alpha_k)\nabla \tilde{U}_k(\theta_k) + \sqrt{2}(\mathcal{W}_s^{(d)} - \mathcal{W}_{\sum_{k=1}^{j} \alpha_k}^{(d)})$$

$$= -(s - \sum_{k=1}^{j} \alpha_k)\nabla U(\theta_k) + (s - \sum_{k=1}^{j} \alpha_k)(\nabla U(\theta_k) - \nabla \tilde{U}_k(\theta_k)) + \sqrt{2}(\mathcal{W}_s^{(d)} - \mathcal{W}_{\sum_{k=1}^{j} \alpha_k}^{(d)}) \tag{16}$$

Thus, we can use Lemma 3.1 and 3.2 in Raginsky et al. (2017) for the following result:

$$\mathbb{E}\|\underline{\theta}(s) - \underline{\theta}(\sum_{k=1}^{j} \alpha_k)\|^2 \leq 3\alpha_{j+1}^2 \mathbb{E}\|\nabla U(\theta_j)\|^2 + 3\alpha_{j+1}^2 \mathbb{E}\|\nabla U(\theta_j) - \nabla \tilde{U}_j(\theta_j)\|^2 + 6\alpha_{j+1}d$$

$$\leq 12\alpha_{j+1}^2(L_U^2 \mathbb{E}\|\theta_j\|^2 + B^2) + 6\alpha_{j+1}d$$

Hence we can bound the first part, (choosing $\alpha_0 \leq 1$),

$$\frac{L_U^2}{2} \sum_{j=0}^{K-1} \int_{\sum_{k=1}^{j} \alpha_k}^{\sum_{k=1}^{j+1} \alpha_k} \mathbb{E}\|\underline{\theta}(s) - \underline{\theta}(\sum_{k=1}^{q(s)} \alpha_i)\|^2 \mathrm{d}s$$

$$\leq \frac{L_U^2}{2} \sum_{j=0}^{K-1} \left[ 12\alpha_{j+1}^3(L_U^2 \mathbb{E}\|\theta_j\|^2 + B^2) + 6\alpha_{j+1}^2 d \right]$$

$$\leq L_U^2 \max_{0 \leq j \leq K-1} \left[ 6(L_U^2 \mathbb{E}\|\theta_j\|^2 + B^2) + 3d \right] (\sum_{j=0}^{K-1} \alpha_{j+1}^2)$$

$$\leq L_U^2 \max_{0 \leq j \leq K-1} \left[ 6(L_U^2 \mathbb{E}\|\theta_j\|^2 + B^2) + 3d \right] \frac{3\alpha_0^2 K}{8} \tag{17}$$

The last line (17) follows from[2] equation 6. The second part (15) can be bounded as follows:

$$\frac{1}{2}\sum_{j=0}^{K-1}\int_{\sum_{k=1}^{j}\alpha_k}^{\sum_{k=1}^{j+1}\alpha_k}\mathbb{E}\|\nabla U(\underline{\theta}(\sum_{k=1}^{q(s)}\alpha_i)-G_s(\underline{\theta}(\sum_{k=1}^{q(s)}\alpha_i))\|^2\mathrm{d}s$$

$$=\frac{1}{2}\sum_{j=0}^{K-1}\alpha_{j+1}\mathbb{E}\|\nabla U(\theta_j)-\nabla\tilde{U}(\theta_j)\|^2$$

$$\leq\sigma\max_{0\leq j\leq K-1}(L_U^2\mathbb{E}\|\theta_j\|^2+B^2)\sum_{j=0}^{K-1}\alpha_{j+1}$$

$$\leq\sigma\max_{0\leq j\leq K-1}(L_U^2\mathbb{E}\|\theta_j\|^2+B^2)(\frac{\alpha_0}{2}\sum_{j=0}^{K-1}(\cos(\frac{\pi mod(j,K/M)}{K/M})+1))$$

$$\leq\sigma\max_{0\leq j\leq K-1}(L_U^2\mathbb{E}\|\theta_j\|^2+B^2)(\frac{K\alpha_0}{2})$$

Due to the data-processing inequality for the relative entropy, we have

$$D_{KL}(\mu_K\|\nu_{\sum_{k=1}^{K}\alpha_k})\leq D_{KL}(\mathbf{P}_V^t\,\|\,\mathbf{P}_\theta^t)$$

$$\leq\frac{L_U^2}{2}\sum_{j=0}^{K-1}\int_{\sum_{k=1}^{j}\alpha_k}^{\sum_{k=1}^{j+1}\alpha_k}\mathbb{E}\|\underline{\theta}(s)-\underline{\theta}(\sum_{k=1}^{q(s)}\alpha_i)\|^2\mathrm{d}s$$

$$+\frac{1}{2}\sum_{j=0}^{K-1}\int_{\sum_{k=1}^{j}\alpha_k}^{\sum_{k=1}^{j+1}\alpha_k}\mathbb{E}\|\nabla U(\underline{\theta}(\sum_{k=1}^{q(s)}\alpha_i)-G_s(\underline{\theta}(\sum_{k=1}^{q(s)}\alpha_i))\|^2\mathrm{d}s$$

$$\leq L_U^2\max_{0\leq j\leq K-1}\left[6(L_U^2\mathbb{E}\|\theta_j\|^2+B^2)+3d\right]\frac{3\alpha_0^2K}{8}$$

$$+\sigma\max_{0\leq j\leq K-1}(L_U^2\mathbb{E}\|\theta_j\|^2+B^2)(\frac{K\alpha_0}{2})$$

According to the proof of Lemma 3.2 in Raginsky et al. (2017), we can bound the term $\mathbb{E}\|\theta_k\|^2$

$$\mathbb{E}\|\theta_{k+1}\|^2\leq(1-2\alpha_{k+1}m_U+4\alpha_{k+1}^2M_U^2)\mathbb{E}\|\theta_k\|^2+2\alpha_{k+1}b+4\alpha_{k+1}^2B^2+\frac{2\alpha_{k+1}d}{\beta}$$

Similar to the statement of Lemma 3.2 in Raginsky et al. (2017), we can fix $\alpha_0\in(0,1\wedge\frac{m_U}{4M_U^2})$. Then, we can know that

$$\mathbb{E}\|\theta_{k+1}\|^2\leq(1-2\alpha_{min}m_U+4\alpha_{min}^2M_U^2)\mathbb{E}\|\theta_k\|^2+2\alpha_0b+4\alpha_0^2B^2+\frac{2\alpha_0d}{\beta}\qquad(18)$$

, where $\alpha_{min}$ is defined as $\alpha_{min}\triangleq\frac{\alpha_0}{2}\left[\cos\left(\frac{\pi\,\mathrm{mod}(\lceil K/M\rceil-1,\lceil K/M\rceil)}{\lceil K/M\rceil}\right)+1\right]$.

There are two cases to consider.

- If $1-2\alpha_{min}m_U+4\alpha_{min}^2M_U^2\leq0$, then from equation 18 it follows that

$$\mathbb{E}\|\theta_{k+1}\|^2\leq2\alpha_0b+4\alpha_0^2B^2+\frac{2\alpha_0d}{\beta}$$

$$\leq\mathbb{E}\|\theta_0\|^2+2(b+2B^2+\frac{d}{\beta})$$

- If $0\leq1-2\alpha_{min}m_U+4\alpha_{min}^2M_U^2\leq1$, then iterating equation 18 gives

$$\mathbb{E}\|\theta_k\|^2\leq(1-2\alpha_{min}m_U+4\alpha_{min}^2M_U^2)^k\mathbb{E}\|\theta_0\|^2+\frac{\alpha_0b+2\alpha_0^2B^2+\frac{\alpha_0d}{\beta}}{\alpha_{min}m_U-2\alpha_{min}^2M_U^2}\qquad(19)$$

$$\leq\mathbb{E}\|\theta_0\|^2+\frac{2\alpha_0}{m_U\alpha_{min}}(b+2B^2+\frac{d}{\beta})\qquad(20)$$

---

[2]Note: we only focus on the case when $K\mod M=0$.

Now, we have

$$\max_{0 \leq j \leq K-1}(L_U^2 \mathbb{E}\|\theta_j\|^2 + B^2)$$

$$\leq (L_U^2(\kappa_0 + 2(1 \wedge \frac{\alpha_0}{m_U \alpha_{min}})(b + 2B^2 + d)) + B^2) := C_0$$

Due to the expression of $\frac{\alpha_0}{\alpha_{min}}$, $C_0$ is independent of $\alpha_0$. Then we denote the $6L_U^2(C_0 + d)$ as $C_1$ and we can derive

$$D_{KL}(\mu_K\|\nu_{\sum_{k=1}^K \alpha_k}) \leq C_1(\frac{3\alpha_0^2 K}{8}) + \sigma C_0(\frac{K\alpha_0}{2})$$

Then according to Proposition 3.1 in Bolley & Villani (2005) and Lemma 3.3 in Raginsky et al. (2017), if we denote $\kappa_0 + 2b + 2d$ as $C_2$, we can derive the following result:

$$W_2(\mu_K, \nu_{\sum_{k=1}^K \alpha_k}) \leq (12 + C_2(\sum_{k=1}^K \alpha_k))^{\frac{1}{2}} \cdot [D_{KL}(\mu_K\|\nu_{\sum_{k=1}^K \alpha_k})^{\frac{1}{2}} + D_{KL}(\mu_K\|\nu_{\sum_{k=1}^K \alpha_k})^{\frac{1}{4}}]$$

$$\leq (12 + \frac{C_2 K\alpha_0}{2})^{\frac{1}{2}} \cdot [(\frac{3C_1\alpha_0^2 K}{8} + \frac{K\sigma C_0\alpha_0}{2})^{\frac{1}{2}} + (\frac{3C_1\alpha_0^2 K}{16} + \frac{K\sigma C_0\alpha_0}{4})^{\frac{1}{4}}]$$

## D.2 $W_2(\nu_{\sum_{k=1}^K \alpha_k}, \nu_\infty)$

We can directly get the following results from (3.17) in Raginsky et al. (2017) that there exist some positive constants $(C_3, C_4)$,

$$W_2(\nu_{\sum_{k=1}^K \alpha_k}, \nu_\infty) \leq C_3 \exp(-\sum_{k=1}^K \alpha_k/C_4)$$

Now combining the bounds for $W_2(\mu_K, \nu_{\sum_{k=1}^K \alpha_k})$ and $W_2(\nu_{\sum_{k=1}^K \alpha_k}, \nu_\infty)$, substituting $\alpha_0 = O(1/K^\beta)$, and noting $W_2(\nu_{\sum_{k=1}^K \alpha_k}, \nu_\infty)$ decreases w.r.t. $K$, we arrive at the bound stated in the theorem.

$\square$

## E RELATION WITH SGLD

For the standard polynomially-decay-stepsize SGLD, the convergence rate is bounded as

$$W_2(\tilde{\mu}_K, \nu_\infty) \leq W_2(\tilde{\mu}_K, \nu_{\sum_{k=1}^K h_k}) + W_2(\nu_{\sum_{k=1}^K h_k}, \nu_\infty) \tag{21}$$

where $W_2(\tilde{\mu}_K, \nu_{\sum_{k=1}^K h_k}) \leq (6 + h_0 \sum_{k=1}^K \frac{1}{k})^{\frac{1}{2}} \cdot$

$$[(D_1 h_0^2 \frac{\pi^2}{6} + \sigma D_0 h_0 \sum_{k=1}^K \frac{1}{k})^{\frac{1}{2}} + (D_1 h_0^2 \frac{\pi^2}{16} + \sigma D_0 \frac{h_0}{2} \sum_{k=1}^K \frac{1}{k})^{\frac{1}{4}}]$$

and $W_2(\nu_{\sum_{k=1}^K h_k}, \nu_\infty) \leq C_3 \exp(-\frac{\sum_{k=1}^K h_k}{C_4})$.

*Proof of the bound of $W_2(\tilde{\mu}_K, \nu_\infty)$ in the standard SGLD.* Similar to the proof of $W_2(\mu_K, \nu_\infty)$ in cSGLD, we get the following update rule for SGLD with the stepsize following a polynomial decay i.e., $h_k = \frac{h_0}{k}$,

$$\theta_{k+1} = \theta_k - \nabla \tilde{U}_k(\theta_k)h_{k+1} + \sqrt{2h_{k+1}}\xi_{k+1} \tag{22}$$

Let $\tilde{\mu}_k$ denote the distribution of $\theta_k$.

Since

$$W_2(\tilde{\mu}_K, \nu_\infty) \leq W_2(\tilde{\mu}_K, \nu_{\sum_{k=1}^K h_k}) + W_2(\nu_{\sum_{k=1}^K h_k}, \nu_\infty) \tag{23}$$

, we need to give the bounds for these two parts respectively.

## E.1 $\quad W_2(\tilde{\mu}_K, \nu_{\sum_{k=1}^K h_k})$

We first assume $\mathbb{E}(\nabla \tilde{U}(w)) = \nabla U(w), \quad \forall w \in \mathbb{R}^d$ , which is a general assumption according to the way we choose the minibatch. Following the proof in Raginsky et al. (2017) and the analysis of the SPOS method in Zhang et al. (2018), we define the following $p(t)$ which will be used in the following proof:

$$p(t) = \{k \in \mathbb{Z} | \sum_{i=1}^k h_i \le t < \sum_{i=1}^{k+1} h_i\} \tag{24}$$

Then we focus on the following continuous-time interpolation of $\theta_k$:

$$\underline{\theta}(t) = \theta_0 - \int_0^t \nabla \tilde{U} \left( \underline{\theta}(\sum_{k=1}^{p(s)} h_k) \right) \mathrm{d}s + \sqrt{2} \int_0^t d\mathcal{W}_s^{(d)}, \tag{25}$$

$$\tag{26}$$

where $\nabla \tilde{U} \equiv \nabla \tilde{U}_k$ for $t \in \left[ \sum_{i=1}^k h_i, \sum_{i=1}^{k+1} h_i \right)$. And for each $k$ , $\underline{\theta}(\sum_{i=1}^k h_i)$ and $\theta_k$ have the same probability law $\tilde{\mu}_k$.
Since $\underline{\theta}(t)$ is not a Markov process, we define the following process which has the same one-time marginals as $\underline{\theta}(t)$

$$V(t) = \theta_0 - \int_0^t G_s \left( V(s) \right) \mathrm{d}s + \sqrt{2} \int_0^t d\mathcal{W}_s^{(d)} \tag{27}$$

with

$$G_t(x) := \mathbb{E} \left[ \nabla \tilde{U} \left( \underline{\theta}(\sum_{i=1}^{q(t)} h_i) \right) | \underline{\theta}(t) = x \right] \tag{28}$$

Let $\mathbf{P}_V^t := \mathcal{L}(V(s) : 0 \le s \le t)$ and $\mathbf{P}_\theta^t := \mathcal{L}(\theta(s) : 0 \le s \le t)$ and according to the proof of Lemma 3.6 in Raginsky et al. (2017), we can derive the similar result for the relative entropy of $\mathbf{P}_V^t$ and $\mathbf{P}_\theta^t$:

$$
\begin{aligned}
D_{KL}(\mathbf{P}_V^t \| \mathbf{P}_\theta^t) &= -\int \mathrm{d}\mathbf{P}_V^t \log \frac{\mathrm{d}\mathbf{P}_V^t}{\mathrm{d}\mathbf{P}_\theta^t} \\
&= \frac{1}{4} \int_0^t \mathbb{E}\|\nabla U(V(s)) - G_s(V(s))\|^2 \mathrm{d}s \\
&= \frac{1}{4} \int_0^t \mathbb{E}\|\nabla U(\underline{\theta}(s)) - G_s(\underline{\theta}(s))\|^2 \mathrm{d}s
\end{aligned}
$$

The last line follows the fact that $\mathcal{L}(\underline{\theta}(s)) = \mathcal{L}(V(s))$, $\forall s$.

Then we will let $t = \sum_{k=1}^{K} h_k$ and we can use the martingale property of integral to derive:

$$D_{KL}(\mathbf{P}_V^{\sum_{k=1}^{K} h_k} \,\|\, \mathbf{P}_\theta^{\sum_{k=1}^{K} h_k})$$

$$= \frac{1}{4} \sum_{j=0}^{K-1} \int_{\sum_{k=1}^{j} h_k}^{\sum_{k=1}^{j+1} h_k} \mathbb{E}\|\nabla U(\underline{\theta}(s)) - G_s(\underline{\theta}(s))\|^2 \mathrm{d}s$$

$$\leq \frac{1}{2} \sum_{j=0}^{K-1} \int_{\sum_{k=1}^{j} h_k}^{\sum_{k=1}^{j+1} h_k} \mathbb{E}\|\nabla U(\underline{\theta}(s)) - \nabla U(\underline{\theta}(\overset{q(s)}{\underset{k=1}{\sum}} h_i))\|^2 \mathrm{d}s$$

$$+ \frac{1}{2} \sum_{j=0}^{K-1} \int_{\sum_{k=1}^{j} h_k}^{\sum_{k=1}^{j+1} h_k} \mathbb{E}\|\nabla U(\underline{\theta}(\overset{q(s)}{\underset{k=1}{\sum}} h_i)) - G_s(\underline{\theta}(\overset{q(s)}{\underset{k=1}{\sum}} h_i))\|^2 \mathrm{d}s$$

$$\leq \frac{L_U^2}{2} \sum_{j=0}^{K-1} \int_{\sum_{k=1}^{j} h_k}^{\sum_{k=1}^{j+1} h_k} \mathbb{E}\|\underline{\theta}(s) - \underline{\theta}(\overset{q(s)}{\underset{k=1}{\sum}} h_i)\|^2 \mathrm{d}s \qquad (29)$$

$$+ \frac{1}{2} \sum_{j=0}^{K-1} \int_{\sum_{k=1}^{j} h_k}^{\sum_{k=1}^{j+1} h_k} \mathbb{E}\|\nabla U(\underline{\theta}(\overset{q(s)}{\underset{k=1}{\sum}} h_i)) - G_s(\underline{\theta}(\overset{q(s)}{\underset{k=1}{\sum}} h_i))\|^2 \mathrm{d}s \qquad (30)$$

For the first part (29), we consider some $s \in [\sum_{k=1}^{j} h_k, \sum_{k=1}^{j+1} h_k)$, the following equation holds:

$$\underline{\theta}(s) - \underline{\theta}(\sum_{k=1}^{j} h_k)$$

$$= -(s - \sum_{k=1}^{j} h_k)\nabla \tilde{U}_k(\theta_k) + \sqrt{2}(\mathcal{W}_s^{(d)} - \mathcal{W}_{\sum_{k=1}^{j} h_k}^{(d)})$$

$$= -(s - \sum_{k=1}^{j} h_k)\nabla U(\theta_k) + (s - \sum_{k=1}^{j} h_k)(\nabla U(\theta_k) - \nabla \tilde{U}_k(\theta_k)) + \sqrt{2}(\mathcal{W}_s^{(d)} - \mathcal{W}_{\sum_{k=1}^{j} h_k}^{(d)}) \qquad (31)$$

Thus, we can use Lemma 3.1 and 3.2 in Raginsky et al. (2017) for the following result:

$$\mathbb{E}\|\underline{\theta}(s) - \underline{\theta}(\sum_{k=1}^{j} h_k)\|^2$$

$$\leq 3h_{j+1}^2 \mathbb{E}\|\nabla U(\theta_j)\|^2 + 3h_{j+1}^2 \mathbb{E}\|\nabla U(\theta_j) - \nabla \tilde{U}_j(\theta_j)\|^2 + 6h_{j+1}d$$

$$\leq 12h_{j+1}^2(L_U^2 \mathbb{E}\|\theta_j\|^2 + B^2) + 6h_{j+1}d$$

Hence we can bound the first part, (choosing $h_0 \leq 1$),

$$\frac{L_U^2}{2} \sum_{j=0}^{K-1} \int_{\sum_{k=1}^{j} h_k}^{\sum_{k=1}^{j+1} h_k} \mathbb{E}\|\underline{\theta}(s) - \underline{\theta}(\overset{q(s)}{\underset{k=1}{\sum}} h_k)\|^2 \mathrm{d}s$$

$$\leq \frac{L_U^2}{2} \sum_{j=0}^{K-1} \left[12h_{j+1}^3(L_U^2 \mathbb{E}\|\theta_j\|^2 + B^2) + 6h_{j+1}^2 d\right]$$

$$\leq L_U^2 \max_{0 \leq j \leq K-1} \left[6(L_U^2 \mathbb{E}\|\theta_j\|^2 + B^2) + 3d\right] (\sum_{j=0}^{K-1} h_{j+1}^2)$$

$$\leq L_U^2 \max_{0 \leq j \leq K-1} \left[6(L_U^2 \mathbb{E}\|\theta_j\|^2 + B^2) + 3d\right] \frac{\pi^2}{6} h_0^2 \qquad (32)$$

where the last line follows from the fact that

$$\sum_{j=0}^{K-1} \frac{1}{(j+1)^3} \leq \sum_{j=0}^{K-1} \frac{1}{(j+1)^2} \leq \sum_{j=0}^{\infty} \frac{1}{(j+1)^2} = \frac{\pi^2}{6}\,.$$

The second part (30) can be bounded as follows:

$$\frac{1}{2}\sum_{j=0}^{K-1}\int_{\sum_{k=1}^{j}h_k}^{\sum_{k=1}^{j+1}h_k}\mathbb{E}\|\nabla U(\underline{\theta}(\sum_{k=1}^{q(s)}h_i)) - G_s(\underline{\theta}(\sum_{k=1}^{q(s)}h_i))\|^2\mathrm{d}s$$

$$=\frac{1}{2}\sum_{j=0}^{K-1}h_{j+1}\mathbb{E}\|\nabla U(\theta_j)-\nabla\tilde{U}(\theta_j)\|^2$$

$$\leq \sigma\max_{0\leq j\leq K-1}(L_U^2\mathbb{E}\|\theta_j\|^2+B^2)\sum_{j=0}^{K-1}h_{j+1}$$

$$\leq \sigma\max_{0\leq j\leq K-1}(L_U^2\mathbb{E}\|\theta_j\|^2+B^2)(h_0\sum_{j=1}^{K}\frac{1}{j})$$

Due to the data-processing inequality for the relative entropy, we have

$$D_{KL}(\tilde{\mu}_K\|\nu_{\sum_{k=1}^{K}h_k}) \leq D_{KL}(\mathbf{P}_V^t\|\mathbf{P}_\theta^t)$$

$$\leq \frac{L_U^2}{2}\sum_{j=0}^{K-1}\int_{\sum_{k=1}^{j}h_k}^{\sum_{k=1}^{j+1}h_k}\mathbb{E}\|\underline{\theta}(s)-\underline{\theta}(\sum_{k=1}^{q(s)}h_i)\|^2\mathrm{d}s$$

$$+\frac{1}{2}\sum_{j=0}^{K-1}\int_{\sum_{k=1}^{j}h_k}^{\sum_{k=1}^{j+1}h_k}\mathbb{E}\|\nabla U(\underline{\theta}(\sum_{k=1}^{q(s)}h_i))-G_s(\underline{\theta}(\sum_{k=1}^{q(s)}h_i))\|^2\mathrm{d}s$$

$$\leq L_U^2\max_{0\leq j\leq K-1}\left[6(L_U^2\mathbb{E}\|\theta_j\|^2+B^2)+3d\right]\frac{\pi^2}{6}h_0^2$$

$$+\sigma\max_{0\leq j\leq K-1}(L_U^2\mathbb{E}\|\theta_j\|^2+B^2)(h_0\sum_{j=1}^{K}\frac{1}{j})$$

Similar to the proof of cSGLD , we have

$$\max_{0\leq j\leq K-1}(L_U^2\mathbb{E}\|\theta_j\|^2+B^2)\leq D_0$$

Then we denote the $6L_U^2(D_0+d)$ as $D_1$ and we can derive

$$D_{KL}(\tilde{\mu}_K\|\nu_{\sum_{k=1}^{K}h_k})\leq D_1h_0^2\frac{\pi^2}{6}+\sigma D_0h_0\sum_{j=1}^{K}\frac{1}{j}$$

Then according to Proposition 3.1 in Bolley & Villani (2005) and Lemma 3.3 in Raginsky et al. (2017), if we denote $\kappa_0+2b+2d$ as $D_2$, we can derive the following result,

$$W_2(\tilde{\mu}_K,\nu_{\sum_{k=1}^{K}h_k})$$

$$\leq [12+D_2(\sum_{k=1}^{K}h_k)]^{1/2}\cdot[(D_{KL}(\tilde{\mu}_K\|\nu_{\sum_{k=1}^{K}h_k}))^{1/2}+(D_{KL}(\tilde{\mu}_K\|\nu_{\sum_{k=1}^{K}h_k})/2)^{1/4}]$$

$$=[12+D_2(h_0\sum_{j=1}^{K}\frac{1}{j})]^{1/2}\cdot[(D_1h_0^2\frac{\pi^2}{6}+\sigma D_0h_0\sum_{j=1}^{K}\frac{1}{j})^{1/2}+(D_1h_0^2\frac{\pi^2}{12}+\sigma D_0h_0\sum_{j=1}^{K}\frac{1}{2j})^{1/4}]$$

Now we derive the bound for $W_2(\tilde{\mu}_K,\nu_{\sum_{k=1}^{K}h_k})$.

## E.2 $W_2(\nu_{\sum_{k=1}^{K}h_k},\nu_\infty)$

We can directly get the following results from (3.17) in Raginsky et al. (2017) that there exist some positive constants $(C_3,C_4)$,

$$W_2(\nu_{\sum_{k=1}^{K}h_k},\nu_\infty)\leq C_3\exp(-\sum_{k=1}^{K}h_k/C_4)$$

$\square$

Based on the convergence error bounds, we discuss an informal comparison with standard SGLD. Consider the following two cases. We must emphasize that since the term $W_2(\mu_K, \nu_{\sum_{k=1}^K \alpha_k})$ in the equation 9 increases w.r.t. $K$, our $\alpha_0$ must be set small enough in practice. Hence, in this informal comparison, we also set $\alpha_0$ small enough to make $W_2(\mu_K, \nu_{\sum_{k=1}^K \alpha_k})$ less important.

*i*) If the initial stepsizes satisfy $\alpha_0 \geq h_0$, our algorithm cSGLD runs much faster than the standard SGLD in terms of the amount of "diffusion time" *i.e.*, the "t" indexing $\theta_t$ in the continuous-time SDE mentioned above. This result follows from $\sum_{k=1}^K \alpha_k = \frac{K\alpha_0}{2}$ and $\sum_{k=1}^K h_k = \sum_{k=1}^K \frac{h_0}{k} = \mathcal{O}(h_0 \log K) \ll \frac{K\alpha_0}{2}$. In standard SGLD, since the error described by $W_2(\tilde{\mu}_K, \nu_{\sum_{k=1}^K h_k})$ increases w.r.t. $K$, $h_0$ needs to be set small enough in practice to reduce the error. Following the general analysis of SGLD in Raginsky et al. (2017); Xu et al. (2017), the dominant term in the decomposition equation 21 will be $W_2(\nu_{\sum_{k=1}^K h_k}, \nu_\infty)$ since it decreases exponentially fast with the increase of $t$ and $W_2(\tilde{\mu}_K, \nu_{\sum_{k=1}^K h_k})$ is small due to the setting of small $h_0$. Since $\sum_{k=1}^K \alpha_k$ increases much faster in our algorithm than the term $\sum_{k=1}^K h_k$ in standard SGLD, our algorithm thus endows less error for K iterations, *i.e.*, $W_2(\nu_{\sum_{k=1}^K \alpha_k}, \nu_\infty) \ll W_2(\nu_{\sum_{k=1}^K h_k}, \nu_\infty)$. Hence, our algorithm outperforms standard SGLD, as will be verified in our experiments.

*ii*) Instead of setting the $h_0$ small enough, one may consider increasing $h_0$ to make standard SGLD run as "fast" as our proposed algorithm, *i.e.*, $\sum_{k=1}^K h_k \approx \sum_{k=1}^K \alpha_k$. Now the $W_2(\nu_{\sum_{k=1}^K h_k}, \nu_\infty)$ in equation 21 is almost the same as the $W_2(\nu_{\sum_{k=1}^K h_k}, \nu_\infty)$ in equation 9. However, in this case, it is worth noting that $h_0$ scales as $\mathcal{O}(\alpha_0 K / \log K)$. We can notice that $h_0$ is much larger than the $\alpha_0$ and thus the $W_2(\tilde{\mu}_K, \nu_{\sum_{k=1}^K h_k})$ cannot be ignored. Now the $h_0^2$ term in $W_2(\tilde{\mu}_K, \nu_{\sum_{k=1}^K h_k})$ would scale as $\mathcal{O}(\alpha_0^2 K^2 / \log^2 K)$, which makes $W_2(\tilde{\mu}_K, \nu_{\sum_{k=1}^K \alpha_k})$ in equation 21 much larger than our $W_2(\mu_K, \nu_{\sum_{k=1}^K \alpha_k})$ defined in equation 9 since $\mathcal{O}(\alpha_0^2 K^2 / \log^2 K) \gg \mathcal{O}(\alpha_0^2 K)$. Again, our algorithm cSGLD achieves a faster convergence rate than standard SGLD.

## F  COMBINING SAMPLES

In cyclical SG-MCMC, we obtain samples from multiple modes of a posterior distribution by running the cyclical step size schedule for many periods. We now show how to effectively utilize the collected samples. We consider each cycle exploring different part of the target distribution $p(\theta|\mathcal{D})$ on a metric space $\Theta$. As we have $M$ cycles in total, the $m$th cycle characterizes a local region $\Theta_m \subset \Theta$, defining the "sub-posterior" distribution: $p_m(\theta|\mathcal{D}) = \frac{p(\theta|\mathcal{D})\mathbf{1}_{\Theta_m}}{w_m}$, with $w_m = \int_{\Theta_m} p(\theta|\mathcal{D})d\theta$, where $w_m$ is a normalizing constant. For a testing function $f(\theta)$, we are often interested in its true posterior expectation $\bar{f} = \int f(\theta)p(\theta|\mathcal{D})d\theta$. The sample-based estimation is

$$\hat{f} = \sum_{m=1}^M w_m \hat{f}_m \ \text{ with } \ \hat{f}_m = \frac{1}{K_m} \sum_{j=1}^{K_m} f(\theta_j^{(m)}), \tag{33}$$

where $K_m$ is the number of samples from the $m$th cycle, and $\theta^{(m)} \in \Theta_m$.

The weight for each cycle $w_i$ is estimated using the *harmonic mean* method (Green, 1995; Raftery et al., 2006): $\hat{w}_m \approx [\frac{1}{K_m} \sum_{j=1}^{K_m} \frac{1}{p(\mathcal{D}|\theta_j^{(m)})}]^{-1}$. This approach provides a simple and consistent estimator, where the only additional cost is to traverse the training dataset to evaluate the likelihood $p(\mathcal{D}|\theta_j^{(m)})$ for each sample $\theta_j^{(m)}$. We evaluate the likelihood once off-line and store the result for testing.

If $\Theta_m$ are not disjoint, we can assume new sub-regions $\tilde{\Theta}_m$ which are disjoint and compute the estimator as following

$$\hat{f}_m = \frac{1}{\bar{n}_m} \sum_{m=1}^M \sum_{j=1}^{K_m} f(\theta_m^{(j)}) \mathbf{1}_{\tilde{\Theta}_m}(\theta_j^{(m)})$$

where

$$\overline{n}_m = \sum_{m=1}^{M} \sum_{j=1}^{K_m} \mathbf{1}_{\tilde{\Theta}_m}(\theta_j^{(m)})$$

and $\mathbf{1}_{\tilde{\Theta}_m}(\theta_j^{(m)})$ equals 1 only when $\theta_j^{(m)} \in \tilde{\Theta}_m$. By doing so, our estimator still holds even if $\Theta_m$ are not disjoint.

## G    THEORETICAL ANALYSIS UNDER CONVEX ASSUMPTION

Firstly, we introduce the following SDE

$$d\theta_t = -\nabla U(\theta_t)dt + \sqrt{2}d\mathcal{W}_t , \tag{34}$$

Let $\nu_t$ denote the distribution of $\theta_t$, and the stationary distribution of equation 34 be $p(\theta|\mathcal{D})$, which means $\nu_\infty = p(\theta|\mathcal{D})$.

However, the exact evaluation of the gradient $\nabla U$ is computationally expensive. Hence, we need to adopt noisy evaluations of $\nabla U$. For simplicity, we assume that at any point $\theta_k$, we can observe the value

$$\nabla \tilde{U}_k = \nabla U(\theta_k) + \zeta_k$$

where $\zeta_k : k = 0, 1, 2, ...$ is a sequence of random (noise) vectors. Then the algorithm is defined as:

$$\theta_{k+1} = \theta_k - \alpha_{k+1}\nabla\tilde{U}_k + \sqrt{2\alpha_{k+1}}\xi_{k+1} \tag{35}$$

Further, let $\mu_k$ denote the distribution of $\theta_k$.

Following the existing work in Dalalyan & Karagulyan (2019), we adopt the following standard assumptions summarized in Assumption 5,

**Assumption 5.**

- *For some positive constants m and M, it holds*

$$U(\theta) - U(\theta') - \nabla U(\theta')^T(\theta - \theta') \geq (m/2)\|\theta - \theta'\|_2^2$$
$$\|\nabla U(\theta) - \nabla U(\theta')\|_2 \leq M\|\theta - \theta'\|_2$$

  *for any $\theta, \theta' \in \mathbb{R}^d$*

- *(bounded bias)* $\mathbb{E}[\|\mathbb{E}(\zeta_k|\theta_k)\|_2^2] \leq \delta^2 d$

- *(bounded variance)* $\mathbb{E}[\|\zeta_k - \mathbb{E}(\zeta_k|\theta_k)\|_2^2 \leq \sigma^2 d$

- *(independence of updates)* $\xi_{k+1}$ *in equation 35 is independent of* $(\zeta_1, \zeta_2, ..., \zeta_k)$

### G.1    THEOREM

Under Assumption 5 in the appendix and $\alpha_0 \in (0, \frac{1}{m} \wedge \frac{2}{M})$, if we define the $\alpha_{min}$ as $\frac{\alpha_0}{2}\left[\cos\left(\frac{\pi \bmod(\lceil K/M\rceil - 1, \lceil K/M\rceil)}{\lceil K/M\rceil}\right) + 1\right]$, we can derive the the following bounds.

If $m\alpha_{min} + M\alpha_0 \leq 2$, then $W_2(\mu_{k+1}, \nu_\infty) \leq$

$$(1 - m\alpha_{min})^K W_2(\mu_0, \nu_\infty) + \frac{(1.65M\alpha_0^{3/2} + \alpha_0\delta)d^{1/2}}{m\alpha_{min}} + \frac{\delta^2\alpha_0 d^{1/2}}{1.65M\alpha_0^{1/2} + \delta + \sqrt{m\alpha_{min}}\delta}. \tag{36}$$

If $m\alpha_{min} + M\alpha_0 > 2$, then $W_2(\mu_{k+1}, \nu_\infty) \leq$

$$(1 - (2 - M\alpha_0))^K W_2(\mu_0, \nu_\infty) + \frac{(1.65M\alpha_0^{3/2} + \alpha_0\delta)d^{1/2}}{2 - M\alpha_0} + \frac{\delta^2\alpha_0 d^{1/2}}{1.65M\alpha_0^{1/2} + \delta + \sqrt{2 - M\alpha_0}\delta}, \tag{37}$$

where the $M, m, \delta, \sigma$ are some positive constants defined in Assumption 5

## G.2 PROOF

*Proof.* According to the equation 1, we can find that the stepsize $\alpha_k$ varies from $\alpha_0$ to $\alpha_{min}$, where $\alpha_{min}$ is defined as $\alpha_{min} \triangleq \frac{\alpha_0}{2} \left[ \cos \left( \frac{\pi \mod(\lceil K/M \rceil - 1, \lceil K/M \rceil)}{\lceil K/M \rceil} \right) + 1 \right]$. When $0 < \alpha_0 < \min(2/M, 1/m)$, it is easy for us to know that $0 < \alpha_k < \min(2/M, 1/m)$ for every $k > 0$. Then we can derive that all the $\rho_k \triangleq \max(1 - m\alpha_k, M\alpha_k - 1)$ will satisfy $0 < \rho_k < 1$. Now according to the Proposition 2 in Dalalyan & Karagulyan (2019), we can derive the result that

$$W_2(\mu_k + 1, \nu_\infty)^2 \leq \{\rho_{k+1} W_2(\mu_k, \nu_\infty) + 1.65 M(\alpha_{k+1}^3 d)^{1/2} + \alpha_{k+1}\delta\sqrt{p}\}^2 + \delta^2 \alpha_{k+1}^2 d \quad (38)$$

Then we will use another lemma derived from Dalalyan & Karagulyan (2019).

**Lemma 2.** *If A,B,C are non-negative numbers such that $A \in (0,1)$ and the sequence of non-negative numbers $y_k$ satisfies the following inequality*

$$y_{k+1}^2 \leq [(1 - A)y_k + C]^2 + B^2$$

*for every integer $k > 0$. Then,*

$$y_k \leq (1 - A)^k y_0 + \frac{C}{A} + \frac{B^2}{C + \sqrt{A}B}$$

Using Lemma 2, we can finish our proof now.

- If $m\alpha_{min} + M\alpha_0 \leq 2$, the $\rho_k$ will satisfy $\rho_k \leq 1 - m\alpha_{min}$ for every $k > 0$. Then the equation 38 will turn into

$$W_2(\mu_{k+1}, \nu_\infty)^2 \leq \{(1 - m\alpha_{min})W_2(\mu_k, \nu_\infty) + 1.65 M(\alpha_0^3 d)^{1/2} + \alpha_0\delta d^{1/2}\}^2 + (\delta\alpha_0 d^{1/2})^2$$

for every $k > 0$. Then we can set $A = m\alpha_{min}$, $C = 1.65 M(\alpha_0^3 d)^{1/2} + \alpha_0\delta d^{1/2}$, $B = \delta\alpha_0 d^{1/2}$ and we can get the result.

- If $m\alpha_{min} + M\alpha_0 > 2$, the $\rho_k$ will satisfy $\rho_k \leq M\alpha_0 - 1$ for every $k > 0$. Then the equation 38 will turn into

$$W_2(\mu_{k+1}, \nu_\infty)^2 \leq \{[1 - (2 - M\alpha_0)]W_2(\mu_k, \nu_\infty) + 1.65 M(\alpha_0^3 d)^{1/2} + \alpha_0\delta d^{1/2}\}^2 + (\delta\alpha_0 d^{1/2})^2$$

for every $k > 0$. Then we can set $A = 2 - M\alpha_0$, $C = 1.65 M(\alpha_0^3 d)^{1/2} + \alpha_0\delta d^{1/2}$, $B = \delta\alpha_0 d^{1/2}$ and we can get the result.

$\square$

## H FUTURE DIRECTION FOR THE WASSERSTEIN GRADIENT FLOWS

We would like to point out that the convergence theorems developed in the above several sections can be potentially applied to study the convergence of the Wasserstein gradient flows (Santambrogio, 2017), which can be regarded as a continuous-time MCMC (Chen et al., 2018; Liu et al., 2019). The theorems may shed some lights on the stepsize choice of the Wasserstein gradient flows which is less studied in the literature. We leave it as an interesting future work.

## I HYPERPARAMETERS SETTING

### I.1 SENSITIVITY OF HYPERPARAMETERS

Compared to SG-MCMC, there are two additional hyperparameters in Algorithm 1: the number of cycles $M$ and the proportion of exploration stage $\beta$. We now study how sensitive they are when comparing to the parallel MCMC. With the same setup as in Section 5.2, We compare our method with $M$ cycles and $L$ epochs per cycle with running $M$ chains parallel MCMC for $L$ epochs. The training budget is 200 epochs. In Table 2, $M = 4$ and $\beta = 0.8$ on CIFAR-10. We compare cSGLD and parallel SGLD with smaller and larger values of $M$ and $\beta$. In Table 6, we see that the conclusion that cSG-MCMC is better than parallel SG-MCMC holds with different values of $M$ and $\beta$.

## I.2 Hyperparameters Setting in Practice

Given the training budget, there is a trade-off between the number of cycles $M$ and the cycle length. We find that it works well in practice by setting the cycle length such that the model with optimization methods will be close to a mode after running for that length. (e.g. the cycle length for CIFAR-10 is 50 epochs. The model optimized by SGD can achieve about 5% error after 50 epochs which means the model is close but not fully converge to a mode after 50 epochs.) Once the cycle length is fixed, $M$ is fixed. $\beta$ needs tuning for different tasks by cross-validation. Generally, $\beta$ needs to be tuned so that the sampler has enough time to reach a good region before starting sampling.

|  | $M = 2, \beta = 0.8$ | $M = 5, \beta = 0.8$ | $M = 4, \beta = 0.7$ | $M = 4, \beta = 0.9$ |
|---|---|---|---|---|
| cSGLD | 4.27 | 4.33 | 4.08 | 4.34 |
| Parallel SGLD | 5.49 | 7.38 | 6.03 | 6.03 |

Table 6: Comparison of test error (%) between cSG-MCMC and parallel algorithm with varying values of hyperparameters on CIFAR-10.

## J Tempering in Bayesian Neural Networks

Tempering is common in modern Bayesian deep learning, for both variational inference and MCMC approaches (e.g., Li et al., 2016a; Nguyen et al., 2017; Fortunato et al., 2017). In general, tempering reflects the belief that the model capacity is misspecified. This combination of beliefs with data is what shapes the posterior we want to use to form a good predictive distribution.

Although we use the prescribed temperature in pSGLD (Li et al., 2016a) for all neural network experiments in the main text ($T \approx 0.0045$), we here investigate the effect of temperature $T$ on performance. We show negative log-likelihood (NLL) and classification error as a function of temperature on CIFAR-10 and CIFAR-100 using cSGLD with the same setup as in Section 5.2. We consider $T \in [1, 0.5, 0.1, 0.05, 0.01, 0.005, 0]$. Figure 6 and 7 show the results on CIFAR-10 and CIFAR-100, respectively. On CIFAR-10, the best performance is achieved at $T = 0.1$ with NLL 0.1331 and error 4.22%. On CIFAR-100, the best performance is achieved at $T = 0.01$ with NLL 0.7835 and error 20.53%. We find that the optimal temperature is often less than 1. We hypothesize that this result is due to the model misspecification common to neural networks.

Indeed, modern neural networks are particularly overparametrized. Tempering enables one to use a model with similar inductive biases to a modern neural network, but with a more well calibrated capacity (which is especially important when we are doing Bayesian integration instead of optimization). Indeed, we show that by sampling from the tempered posterior, we outperform optimization. Learning the amount of tempering by cross-validation is a principled way of aligning the tempering procedure with representing a reasonable posterior. We have shown that sampling with cSGMCMC with tempering helps in terms of both NLL and accuracy, which indicates that we are finding a better predictive distribution.

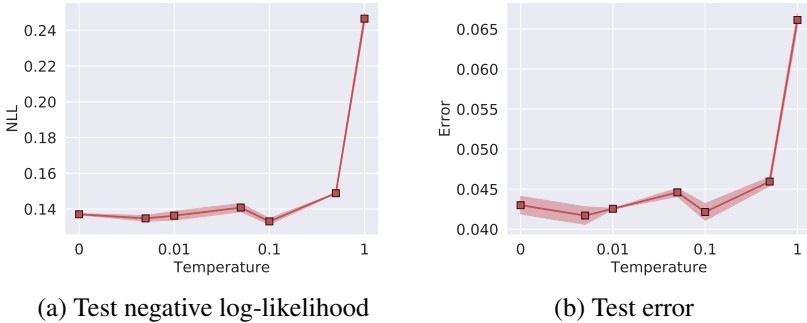

(a) Test negative log-likelihood      (b) Test error

Figure 6: NLL and error (%) as a function of tememprature on CIFAR-10 using cSGLD. The best performance of both NLL and error is achieved at $T = 0.1$.

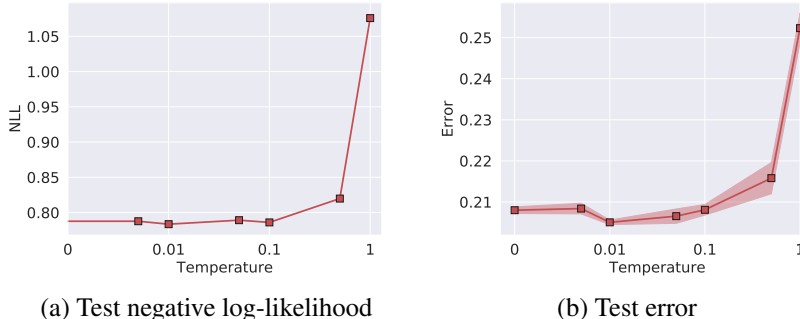

(a) Test negative log-likelihood          (b) Test error

Figure 7: NLL and error (%) as a function of temeprature on CIFAR-100 using cSGLD. The best performance of both NLL and error is achieved at $T = 0.01$.

# K    EXPERIMENTAL SETTING DETAILS

## K.1    BAYESIAN LOGISTIC REGRESSION

For both cSGLD and cSGHMC, $M = 100$, $\beta = 0.01$. For cSGLD, $\alpha_0 N = 1.2, 0.5, 1.5$ for Austrilian, German and Hear respectively. For cSGHMC $\alpha_0 N = 0.5, 0.3, 1.0$ for Austrilian, German and Hear respectively. For SG-MCMC, the stepsize is $a$ for the first 5000 iterations and then switch to the decay schedule (2) with $b = 0$, $\gamma = 0.55$. $aN = 1.2, 0.5, 1.5$ for Austrilian, German and Hear respectively for SGLD and $aN = 0.5, 0.3, 1.0$ for Austrilian, German and Hear respectively for SGHMC. $\eta = 0.5$ in cSGHMC and SGHMC.

Assume that we collect $\{\theta_b\}_{b=1}^B$ samples. *Effective sample size* (ESS) is computed by

$$\text{ESS} = \frac{B}{1 + 2\sum_s^{B-1}(1 - \frac{s}{B})\rho_s}$$

where $\rho_s$ is estimated by

$$\hat{\rho}_s = \frac{1}{\hat{\sigma}^2(B - s)} \sum_{b=s+1}^B (\theta_b - \hat{\mu})(\theta_{b-s} - \hat{\mu})$$

Similar to Hoffman & Gelman (2014), $\hat{\sigma}^2$ and $\hat{\mu}$ are obtained by running an independent sampler. We use HMC in this paper.

## K.2    BAYESIAN NEURAL NETWORKS

For SG-MCMC, the stepsize decays from 0.1 to 0.001 for the first 150 epochs and then switch to the decay schedule (2) with $a = 0.01, b = 0$ and $\gamma = 0.5005$. $\eta = 0.9$ in cSGHMC, Snapshot-SGDM and SGHMC.

## K.3    UNCERTAINTY EVALUATION

For both cSG-MCMC and Snapshot, $M = 4$. $\beta = 0.8$ in cSG-MCMC. $\alpha_0 N = 0.01$ and $0.008$ for cSGLD and cSGHMC respectively. For SG-MCMC, the stepsize is $a$ for the first 50 iterations and then switch to the decay schedule (2) with $b = 0$, $\gamma = 0.5005$. $aN = 0.01$ for SGLD and $aN = 0.008$ for SGHMC. $\eta = 0.5$ in cSGHMC, Snapshot-SGDM and SGHMC.

