# OpenReview forum: "Cyclical Stochastic Gradient MCMC for Bayesian Deep Learning"
_ICLR.cc/2020/Conference — Accept (Talk)_

### Official Review · AnonReviewer1 · 2019-10-23
**Official Blind Review #1**

**Rating:** 8

**Review:**

The paper develops a cyclical stepsize schedule for choosing stepsize for Langevin dynamics.
The authors prove the non-asymptotic convergence theory of the proposed algorithm. Many experimental results, including ImageNet, are given to demonstrate the effectiveness of the proposed method.

Here I suggest that authors also need to point out that the continuous-time MCMC is the Wasserstein gradient flow of KL divergence. The bound derived in this paper focus on the step size choice of gradient flows. This could be a good direction for combining gradient flows studies in optimal transport and MCMC convergence bound for the choice of step size.

Overall, I think that the paper is well written with clear derivations. I strongly suggest the publication of this paper.

**Experience Assessment:**

I have published in this field for several years.

**Review Assessment: Checking Correctness Of Derivations And Theory:**

I assessed the sensibility of the derivations and theory.

**Review Assessment: Checking Correctness Of Experiments:**

I assessed the sensibility of the experiments.

**Review Assessment: Thoroughness In Paper Reading:**

I read the paper at least twice and used my best judgement in assessing the paper.

---

> ### Author Response · Authors · 2019-11-14
> **Response to Reviewer #1**
>
> We appreciate your supportive and thoughtful review.
>
> We agree with your comment on the connection between the continuous-time MCMC and the Wasserstein gradient flows, and think it is really a good idea to combine results from these two fields. For example, as far as we know, there is relatively less theory to describe the convergence behavior of numerical solutions of the Wasserstein gradient flows, whereas convergence of both discrete-time and continuous-time SG-MCMC is fairly well understood. We think it is a good idea to borrow tools from SG-MCMC to study the convergence of the Wasserstein gradient flows. We have added a discussion about the Wasserstein gradient flow in Appendix G.3 in the paper.

---

### Official Review · AnonReviewer3 · 2019-10-24
**Official Blind Review #3**

**Rating:** 8

**Review:**

The paper propose a new MCMC scheme which is demonstrated to perform well for estimating Bayesian neural networks. The key idea is to not keep lowering the step sizes, but -- at pre-specified times -- go back to large step sizes.

The paper is timely, the proposed algorithm is novel, and the theoretical analysis also seem quite novel.

My key concern is that with MCMC sampling it is often quite difficult to tune parameters, and by introducing more parameters to tune when step sizes should increase, I fear that we end up in a "tuning nightmare". How sensitive is the algorithm to choice of parameters?

I would expect that the proposed algorithm is quite similar to just running several MCMCs in parallel. The authors does a comparison to this and show that their approach is significantly faster due to "warm restarts". Here I wonder how sensitive this conclusion is to choice of parameters (see nightmare above) ? I would guess that opposite conclusions could be reached by tuning the algorithms differently -- is that a reasonable suspicion ?

It is argued that the cyclic nature of the algorithms gives a form of "warm start" that is beneficial for MCMC. My intuition dictate that this is only true of the modes of the posterior are reasonable close to each other; otherwise I do not see how this warm starting is helpful. I would appreciate learning more about why this intuition is apparently incorrect.

Minor comments:
* on page 4 it is stated that the proposed algorithm "automatically" provide the warm restarts -- but is it really automatic? Isn't this a priori determined by choice of parameters for the algorithm?

* It would be good to use \citet instead of \cite at places, e.g. "discussed in (Smith & Topin, 2017)" should be "discussed by Smith & Topin (2017)". This would improve readability (which is generally very good).

* For the empirical studies I think it would be good to report state-of-the-art results as well. I expect that the Bayesian nets still are subpar to non-Bayesian methods, and I think the paper should report this.

**Experience Assessment:**

I do not know much about this area.

**Review Assessment: Checking Correctness Of Derivations And Theory:**

I did not assess the derivations or theory.

**Review Assessment: Checking Correctness Of Experiments:**

I assessed the sensibility of the experiments.

**Review Assessment: Thoroughness In Paper Reading:**

I made a quick assessment of this paper.

---

> ### Author Response · Authors · 2019-11-14
> **Response to Reviewer #3**
>
> Thanks for your supportive and thoughtful comments.
>
> Q1: Tuning hyperparameters
>
> A1:  We have found performance robust to hyperparameter settings. There are only two additional hyperparameters compared to standard SG-MCMC methods: the number of cycles $M$ and the proportion of exploration stage $\beta$. Given the training budget, there is a trade-off between the number of cycles $M$ and the cycle length. We find that it works well in practice to set the cycle length such that the model with optimization methods will be close to a mode after running for that length, but it is not too sensitive to this value. (e.g. the cycle length for CIFAR-10 is 50 epochs. The model optimized by SGD can achieve about 5% error after 50 epochs which means the model is close but not fully converged to a mode after 50 epochs.) Once the cycle length is fixed, $M$ is fixed. $\beta$ needs tuning for different tasks by cross-validation. Generally, $\beta$ needs to be tuned so that the sampler has enough time to reach a good region before starting sampling. We have added a discussion about hyperparameter tuning in Appendix. As per below (Q2), we have found performance robust to $M$ and $\beta$.
>
> Q2: How sensitive the conclusion of superiority over parallel MCMC is to the choice of parameters
>
> A2: To test the sensitivity, we added an experiment in Appendix. H. With the same setup as in the experiment section, we compare our method with $M$ cycles and $L$ epochs per cycle with running $M$ chains parallel MCMC for $L$ epochs. The training budget is 200 epochs. We compare cSGLD and parallel SGLD with smaller and larger values of $M$ and $\beta$ on CIFAR-10 ($M=4$ and $\beta=0.8$ in the experiment section). From the results, we see that the conclusion that cSG-MCMC is better than parallel SG-MCMC holds with different values of $M$ and $\beta$.
>
> Table: Comparison of test error (%) between cSG-MCMC and parallel algorithm with varying values of hyperparameters on CIFAR-10.
> 		| $M=2,\beta=0.8$ | $M=5,\beta=0.8$ | $M=4,\beta=0.7$ | $M=4,\beta=0.9$ |
> cSGLD		 |	4.27	      |	4.33	        |	4.08	           |	      4.34	     |
> Parallel SGLD |	5.49	      |	7.38	        |	6.03	           |	      6.03	     |
>
>
> Q3: Is the "warm start" beneficial only when the modes of the posterior are reasonably close to each other?
>
> A3: The loss surfaces for neural nets are extremely complex, containing many modes, mode connecting curves, and a rich variety of high performing solutions even when taking small steps in weight space, which was recently discovered in [1]. In other words, high performing solutions in the weight space of neural networks are likely to be reasonably close to each other. Our method is developed particularly for Bayesian deep learning and we have demonstrated its effectiveness for modeling the posterior in Bayesian neural networks.
>
> Q4: Is the method really automatic?
>
> A4: As in A2, we did not find that our key conclusions are sensitive to the values of hyperparameters. But in order to achieve the best results, it is useful to tune the parameters through simple heuristics and validation, as described above. It is a relatively automatic approach.
>
> Q5: It would be good to use \citet instead of \cite
>
> A5: Thanks for pointing it out. We have revised the paper accordingly.
>
> Q6: Bayesian vs non-Bayesian methods for SOTA
>
> A6: State-of-the-art results often use complex architectures along with many other techniques, often orthogonal to Bayesian vs non-Bayesian statistics, as well as GPU or TPU resources well beyond reach for most academic laboratories, making a direct comparison with our approach difficult to obtain and interpret. In the experiments in this paper, we used the same ResNet architecture for all methods and compare our methods to both related sampling methods and optimization methods (Bayesian and non-Bayesian). Therefore the comparisons are fair, and demonstrate the improvement is due to the proposed cyclical sampling method. We note that our Bayesian approach outperforms the non-Bayesian training methods when controlling for the other factors (e.g. same architecture). Thus the results are a timely step forward for practical approaches to Bayesian deep learning.
>
> [1] Garipov et.al., Loss Surfaces, Mode Connectivity, and Fast Ensembling of DNNs

---

> > ### Comment · AnonReviewer3 · 2019-11-15
> > **Rebuttal acknowledged**
> >
> > Thanks for the detailed reply, which clarified matters for me. I will update my score accordingly.
> >
> > I very much agree that the paper is timely and valuable. I still think it would be good to compare to non-Bayesian SOTA even if it isn't "fair". There is still a (unfortunate) performance gap between Bayesian and non-Bayesian methods, which I think is good to acknowledge. The gap does not diminish the contributions of the paper.

---

### Official Review · AnonReviewer2 · 2019-10-25
**Official Blind Review #2**

**Rating:** 6

**Review:**

This article presents cyclical stochastic gradient MCMC for Bayesian deep learning for inference in posterior distributions of network weights of Bayesian NNs. The posteriors of Bayesian NN weights are highly multi-modal and present difficulty for standard stochastic gradient MCMC methods. The proposed cyclical version periodically warm start the SG-MCMC process such that it can explore the multimodal space more efficiently.

The proposed method as well as the empirical results intuitively make sense. The standard SG-MCMC basically has one longer stepsize schedule and is exploring the weight space more patiently, but only converges to one local mode. The cyclical SG-MCMC uses multiple shorter stepsize schedules, so each one is similar to a (stochastic) greedy search. Consequently, the cSG-MCMC can collect more diverse samples across the weight space, while the samples of SG-MCMC are more concentrated, but likely with better quality (as shown in Figure 3).

Personally I would like to see how Bayesian deep learning can be applied to real large-scale applications. Probabilistic inference is expensive; Bayesian model averaging is even more expensive. That's probably why recent literature focuses on variational inference or expectation propagation-based approaches.

**Experience Assessment:**

I have read many papers in this area.

**Review Assessment: Checking Correctness Of Derivations And Theory:**

I assessed the sensibility of the derivations and theory.

**Review Assessment: Checking Correctness Of Experiments:**

I assessed the sensibility of the experiments.

**Review Assessment: Thoroughness In Paper Reading:**

I read the paper at least twice and used my best judgement in assessing the paper.

---

> ### Author Response · Authors · 2019-11-14
> **Response to Reviewer #2**
>
> Thanks for your supportive and valuable comments.
>
> Q: I would like to see how Bayesian deep learning can be applied to real large-scale applications.
>
> A: A significant benefit of cSGMCMC is its relatively high scalability for Bayesian deep learning. Indeed, we have applied our method to ImageNet in Section 5.2, which is a hard large-scale image classification problem, containing 1000 categories and 1.2 million images for training. Most published work in Bayesian deep learning (sampling or otherwise) does not consider datasets at this scale. We have shown that our method significantly outperforms previous sampling methods and also provides better test likelihood than optimization methods. By exploring the parameter space more efficiently with the cyclical schedule, our method is able to collect more diverse samples and as a result increase the number of effective samples, reducing the number of needed samples and thus the cost of Bayesian averaging.
>
> We agree that recent literature in Bayesian deep learning has been focusing on deterministic approaches such as variational inference. This is why this paper makes an extremely timely and significant contribution to Bayesian deep learning. While MCMC was once a gold standard for inference with neural networks, essentially all inference approaches currently use deterministic approximations for modern deep neural networks. A key advantage of MCMC is the ability to explore complex multimodal distributions. We show in this work for the first time how MCMC can be particularly developed for practical inference over multiple modes in modern deep learning, corresponding to meaningfully different representations of the data, providing complementary advantages to deterministic methods (which are largely unimodal) and classical optimization approaches (which do not perform Bayesian marginalization). We hope you can consider this response, and the foundational and timely nature of this contribution, in your final assessment.

---

### Author Response · Authors · 2019-11-14
**Changes in the New Version**

We have incorporated reviewers’ suggestions and comments into the new version. The changes are the following:

1. Appendix G.3. Future Direction for the Wasserstein gradient flows. We have added a discussion about the relationship between MCMC and the Wasserstein gradient flow.

2. Appendix H. Sensitivity of Hyperparameters. We have added an experiment to test the robustness of the comparison between our method and parallel MCMC. We found that the results in the paper hold over a range of hyperparameters.

3. Appendix I. Tempering in Bayesian Neural Networks. We have added a discussion about tempering in Bayesian neural networks. We found that a tempered posterior is beneficial for Bayesian inference with neural networks and tuning the temperature can further improve the results.

---

### Public Comment · ~Amit_Chandak1 · 2020-01-14
**Missing comparison with other SOTA methods**

Hi,
You have compared  with SnapShot-SGD and not https://github.com/timgaripov/dnn-mode-connectivity which is better than SnapShot-SGD. Also in case of uni-modal distributions it might be better to compare with https://github.com/wjmaddox/swa_gaussian

Thank You

---

### Decision · Program_Chairs · 2019-12-19

**Decision:**

Accept (Talk)

**Comment:**

This paper proposes a novel stochastic gradient Markov chain Monte Carlo method incorporating a cyclical step size schedule (cyclical SG-MCMC).  The authors argue that this step size schedule allows the sampler to cross modes (when the step size is large) and locally explore modes (when the step size is smaller).  SG-MCMC is a very promising method for Bayesian deep learning as it is both scalable and easily to incorporate into existing models.  However, the stochastic setting often leads to the sampler getting stuck in a local mode due to a requirement of a small step size (which itself is often due to leaving out the Metropolis-Hastings accept / reject step).   The cyclic learning rate intuitively helps the sampler escape local modes.  This property is demonstrated on synthetic problems in comparison to existing SG-MCMC baselines.  The authors demonstrate improved negative log likelihood on larger scale deep learning benchmarks, which is appreciated as the related literature often restricts experiments to small scale problems.  The reviewers all found the paper compelling and argued for acceptance and thus the recommendation is to accept.  Some questions remain for future work.  E.g. all experiments were performed using a very low temperature, which implies that the methods are not sampling from the true Bayesian posterior.  Why is such a low temperature needed for reasonable performance?  In any case a very nice paper.